# Fetuin-A is a HIF target that safeguards tissue integrity during hypoxic stress

Stefan Rudloff [1,2], Mathilde Janot[1,2], Stephane Rodriguez[1,2,4], Kevin Dessalle[1,2], Willi Jahnen-Dechent [3] & Uyen Huynh-Do [1,2✉]

Intrauterine growth restriction (IUGR) is associated with reduced kidney size at birth, accelerated renal function decline, and increased risk for chronic kidney and cardiovascular diseases in adults. Precise mechanisms underlying fetal programming of adult diseases remain largely elusive and warrant extensive investigation. Setting up a mouse model of hypoxia-induced IUGR, fetal adaptations at mRNA, protein and cellular levels, and their long-term functional consequences are characterized, using the kidney as a readout. Here, we identify fetuin-A as an evolutionary conserved HIF target gene, and further investigate its role using fetuin-A KO animals and an adult model of ischemia-reperfusion injury. Beyond its role as systemic calcification inhibitor, fetuin-A emerges as a multifaceted protective factor that locally counteracts calcification, modulates macrophage polarization, and attenuates inflammation and fibrosis, thus preserving kidney function. Our study paves the way to therapeutic approaches mitigating mineral stress-induced inflammation and damage, principally applicable to all soft tissues.

[1] Department of Nephrology and Hypertension, Bern University Hospital, Freiburgstrasse 15, 3010 Bern, Switzerland. [2] Department of Biomedical Research, University of Bern, Freiburgstrasse 15, 3010 Bern, Switzerland. [3] Helmholtz-Institute for Biomedical Engineering, Biointerface Laboratory, RWTH Aachen University Medical Faculty, Pauwelsstrasse 30, 52074 Aachen, Germany. [4] Present address: Department of Onco-haematology, Geneva Medical University, Geneva, Switzerland. ✉email: uyen.huynh-do@insel.ch

More than 30 years ago, Barker and colleagues[1] introduced the fetal origin of adult disease theory, proposing that adverse events during pregnancy result in low birthweight (LBW) and increase the risk of ischemic heart disease, stroke, or hypertension in adulthood. Over the last decades, this paradigm has been extended to various disorders, including schizophrenia, diabetes mellitus, obstructive pulmonary disease, or chronic kidney diseases (CKD)[2].

LBW, defined by the WHO as a birthweight below 2500 g regardless of gestational age, is either caused by being small for gestational age (SGA), preterm birth, or a combination thereof. SGA, most commonly a consequence of intrauterine growth restriction (IUGR), is a condition in which a fetus does not reach its genetically pre-determined growth potential. It is defined as a birthweight below the 10th percentile, or more stringent as a weight 2SD below the mean[3]. IUGR is associated with numerous unfavorable gestational conditions, including sociodemographic, genetic, and environmental factors[4]. Among the latter, intrauterine hypoxia plays a prominent role that can stem from multiple aetiologies ranging from umbilical cord obstruction, smoking to preeclampsia, or placental insufficiency[5,6]. Hypoxia activates a tightly controlled signaling cascade by stabilization of the hypoxia-inducible transcription factor (HIF) alpha and transactivation of target genes. HIF targets include metabolic enzymes and cell cycle regulators involved in hypoxia adaptation, as well as excreted proteins, including growth factors, immunomodulators, or extracellular matrix components.

In this study, we set up and validated a mouse model of IUGR by exposing gravid mice to normobaric hypoxia, mimicking high elevation. We chose this approach, because high altitude (HA; >2500 m), at which >2% of the world's population reside, is itself inversely correlated with birthweight[7–9]. This makes pregnancies at high altitude the most important cause of maternofetal hypoxia and IUGR worldwide[10]. The prevailing chronic hypoxic conditions at HA begin to negatively affect intrauterine growth of the human fetus at the end of the second trimester[10], a time window overlapping with the main phase of nephrogenesis. The kidneys of such SGA infants are characterized by a lower number of nephrons at birth. Furthermore, population studies conducted among highland residents of various ethnicities found a significant correlation between high altitude and incidence of renal disease[11], making long-term hypoxia an important factor implicated in CKD and end-stage renal disease[12,13]. Although many theories have been proposed[14], the exact mechanisms whereby reduced nephron endowment drives the onset and progression of CKD remain to be defined.

Here, we show that fetal hypoxia induces an ectopic expression profile of liver-specific genes in fetal kidneys. Of these, we identified the plasma glycoprotein fetuin-A (Ahsg) as an evolutionary conserved HIF target gene, and further investigated its role in hypoxic fetal kidneys using fetuin-A KO mice. Beyond its recognized role as systemic calcification inhibitor, our findings establish fetuin-A as a local calcium mineral scavenger, not only counteracting intrarenal calcification, but also attenuating renal fibrosis and inflammation through TGF-β1 antagonization and regulation of macrophage polarization. Our results provide robust mechanistic evidence to support Barker's hypothesis and highlight potential molecular mechanisms that link prenatal hypoxia-induced IUGR to accelerated renal fibrosis and function decline in the adult. This role of fetuin-A paves the way for therapeutic strategies for mineral stress-induced damage in soft tissues.

## Results

**Chronic fetal hypoxia induces IUGR in mice.** To model chronic fetal hypoxia, timed-mated pregnant mice were exposed to 10% oxygen from E14.5 to E18.5 (Fig. 1a). We observed that dams in hypoxia ate 8.6% less throughout the whole gestation compared to dams under ambient conditions (Fig. 1b). For a better visualization, we plotted the differential food intake between these two groups as percent body weight and calculated the total amount of consumed food as area under the curve (AUC). The reduced food intake took place within the first 48 h of hypoxia, the time period the mice seemed to need to adapt to hypoxia. Thereafter, on hypoxia day 3 and 4, food intake was back at 84% ($p = 0.04$) and 93% (no significant difference), respectively. Since caloric restriction itself is a known inducer of IUGR[5], we included an additional control group in our analysis, this is the "caloric control" (Cc) group, to rule out the possibility that our findings might not be due to hypoxia, but rather to reduced ingested calories. In the Cc group, normoxic dams were fed with an amount of food matching the amount of food consumed by the hypoxic mice (Fig. 1b). Thus, throughout gravidity hypoxic and Cc dams consumed 91.4% of the food eaten by normoxic dams, which is a very mild food restriction compared to the majority of other published protocols[15–17]. During gestation, normoxic dams gained 56% of body weight, while the weight increase for Cc or hypoxic dams was 49% or 33%, respectively (Fig. 1c). In more detail, the 7% difference between normoxic and Cc dams was due to a zero net weight gain of Cc mice during the first 24 h of food restriction, thereafter weight gain normalized or was even higher than for normoxic dams. On the contrary, hypoxic dams lost 9.1% of body weight during the initial 24 h of hypoxia and it took 72 h before their weight gain was back to control levels. Despite these differences, placental mass and the number of E18.5 fetuses per litter were indistinguishable among the three groups (Supplementary Fig. 1a, b). However and importantly, only hypoxic E18.5 fetuses showed LBW, fulfilling small for gestational age (SGA) criteria[18], whereas Cc fetuses did not (Fig. 1d). Calculating the E18.5 fetal weight/maternal weight ratio revealed no significant difference among the groups (Supplementary Fig. 1c). Kidneys of hypoxic fetuses were smaller with significantly fewer nephrons compared to controls (Fig. 1e, Supplementary Fig. 2 and Supplementary Movies 1 (normoxia) and 2 (hypoxia)). Moreover, the number of nephrons/E18.5 fetal weight ratio was 1411 in normoxic vs. 1300 in hypoxic fetuses, which further illustrates that nephrogenesis was severely disturbed in SGA fetuses exposed to chronic hypoxia. Tracking the litter size of hypoxic and normoxic dams confirmed that the numbers of pups per litter were comparable and remained constant throughout the suckling phase (Supplementary Fig. 1d). Cannibalism of pups was an exception in our study. If it occurred, it usually affected only the weakest pup in the afflicted litters in the first few days after birth, regardless of the experimental condition. Thus, no difference was evident in survival assessments (Fig. 1f). Importantly, we found striking differences in the postnatal growth of the offspring in relation to sex, genotype, and experimental condition, including a pronounced catch-up growth of hypoxic pups (Fig. 1g and Supplementary Fig. 1e, f).

**Hypoxic fetal kidneys adopt a hepatic gene expression pattern.** To determine whether our approach indeed induced hypoxic conditions in the fetus, we assessed the mRNA expression level of the classic hypoxia-induced target gene Epo in fetal liver samples. We found that its transcription was almost fourfold higher in hypoxic samples than in normoxic controls (Supplementary Fig. 3a), thus confirming that exposure of dams to chronic hypoxia truly activates the transcription of HIF target genes in the fetus. Next, we examined whole genome expression in fetal hypoxic kidneys using gene arrays. We identified 62 induced and 28 repressed genes compared to both control groups (Fig. 2a and

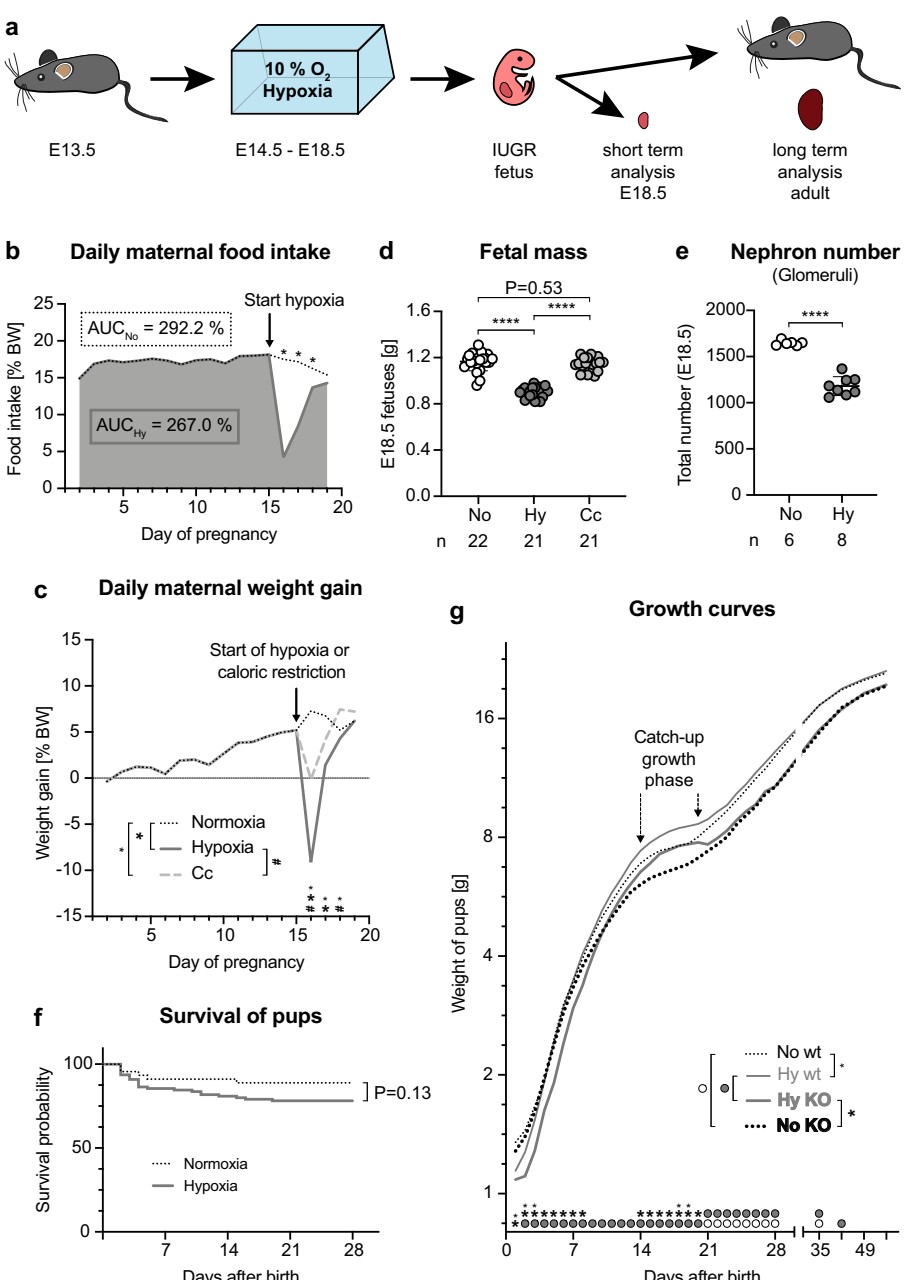

**Fig. 1 Chronic fetal hypoxia induces intrauterine growth restriction in mice. a** Experimental setup and time points of analysis. **b** Mean relative daily food intake shown for all dams until start of hypoxia (arrow), thereafter normoxic dams (black dotted line) or hypoxic dams (bold dark gray line) are separated. During whole gestation, normoxic dams consumed 292.2% of their body weight, while hypoxic mice ate 267%. This corresponds to 91.4% of the food consumed by normoxic mice. Asterisks denote significance (day 16 and 17 $P < 0.0001$, day 18 $P = 0.0398$). **c** Mean relative daily maternal weight gain shown for all dams until the start of intervention (arrow), thereafter normoxic dams (black dotted line), hypoxic dams (bold dark gray line), or Cc dams (dashed light gray line) are separated. Significance is denoted by large asterisks (normoxia vs. hypoxia: day 16 and 17 $P < 0.0001$), small asterisks (normoxia vs. Cc: day 16 $P = 0.0002$, day 17 $P = 0.0012$, day 18 $P = 0.0082$) or # (hypoxia vs. Cc: day 16 $P < 0.0001$, day 18 $P = 0.0007$). **d** Fetal mass shown as mean ± SEM. $N$ = fetuses. Ordinary one-way ANOVA with Tukey's multiple comparison test. **e** Number of nephrons per E18.5 kidney determined by staining for the glomerular marker nephrin shown as mean ± SEM. $N$ = fetal kidneys. Unpaired 2-sided $t$-test with Welch's correction. **f** Survival of normoxic and hypoxic pups. Mantel-Cox log-rank test. **g** Mean postnatal weight of hypoxic offspring (*Ahsg* KO—bold gray line, wild-type-thin gray line) and normoxic offspring (*Ahsg* KO—bold black dotted line, wild-type—thin black dotted line). KOs weighed less than wild-types, catch-up growth was observed for hypoxic offspring during the third week after birth. Gray circles denote significance between hypoxic offspring, white circles between normoxic offspring, large asterisks between KOs, and small asterisks between wild-types. $P$-values are listed in Supplementary Table 8. Individual $P$-values are denoted above the comparison lines (**d**, **e**). $N$ = dams (**b**, **c**) or pups (**f**, **g**) can be derived from Source Data. Multiple 2-sided $t$-tests (**b**, **c**, **g**). (****$P < 0.0001$). Source data are provided as a Source Data file.

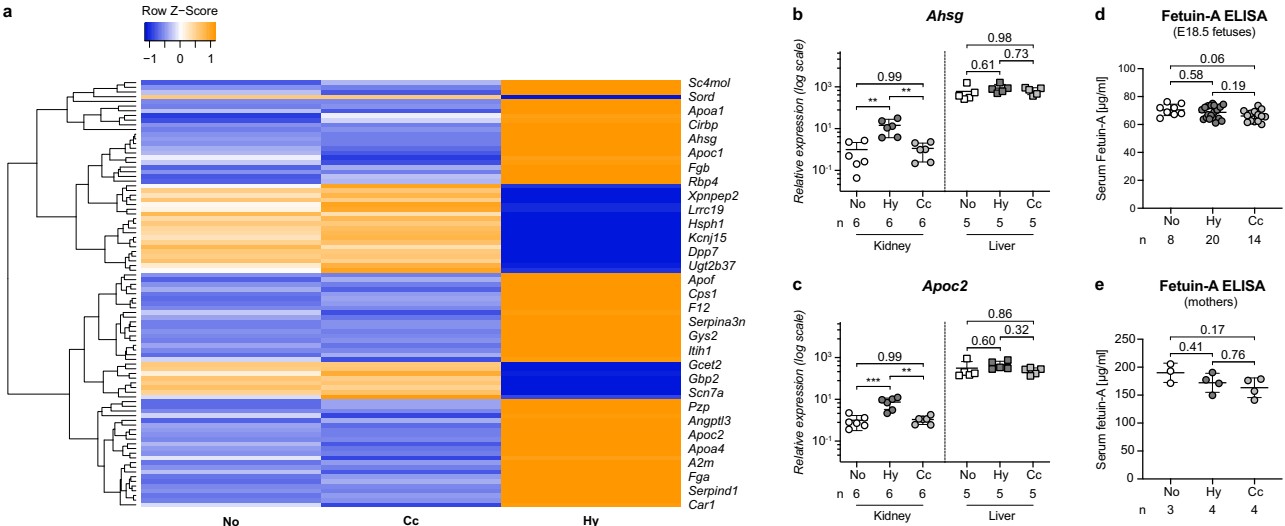

**Fig. 2 Hypoxia-induced gene expression in the kidney. a** Hierarchical clustering of cDNA microarray data comparing the renal gene expression profiles of hypoxic (Hy), normoxic (No), and caloric control (Cc) group E18.5 fetuses ($N = 3$ per experimental condition). Orange indicates induction, blue repression. Clustering was performed for genes with at least 1.3-fold regulation of hypoxic vs. both normoxic controls; 1way ANOVA; $P < 0.05$. **b, c** Relative mRNA values of *Ahsg* (**b**) or *Apoc2* (**c**) in E18.5 kidneys (circles) or liver samples (squares) shown as mean ± SEM. $N$ = fetal organs. Kidney and liver samples are analyzed separately. Note the logarithmic scale on the y-axis. **d, e** Serum fetuin-A levels assessed by ELISA are presented as mean and ± SEM. $N$ = serum samples. No significant changes were observed among normoxic (No), hypoxic (Hy), or caloric control (Cc) E18.5 fetuses (**d**), nor for their mothers (**e**). Ordinary one-way ANOVA with Tukey's multiple comparison test (**b–e**). Individual P-values are denoted above the comparison lines (**b–e**). (***$P < 0.001$; **$P < 0.01$). Source data are provided as a Source Data file.

Supplementary Tables 1 and 2). Of the induced genes, 17 are known to be regulated by hypoxia, including the bona fide HIF target genes transferrin, trefoil factor 3, neuritin, alpha-1-antitrypsin (*Serpina1d*) and alpha-1-antichymotrypsin (*Serpina3n*)[19–22] (Supplementary Table 1). Furthermore, we found more than 20% of the induced genes to be frequently purified from calciprotein particles (CPPs), comprising the major CPP components fetuin-A (*Ahsg*), albumin, Apo-A1 and thrombin (*F2*)[23,24] (Supplementary Table 1). Functional annotation clustering of the induced genes revealed in hypoxic kidneys an enrichment of secreted plasma proteins that are normally transcribed exclusively in the liver. These genes included complement and coagulation factors, proteins involved in lipid metabolism and transport, as well as components of acute phase and acute inflammatory responses (Supplementary Table 3). Validation of the microarray data by quantitative reverse transcription PCR (RT-qPCR) of select genes confirmed a more than fourfold induction in hypoxic fetal kidneys only, but not in control group kidneys nor in the liver (Fig. 2b, c and Supplementary Fig. 3b–f; note the logarithmic scale on the y-axis). Interestingly, *Ahsg*, the gene with the highest induction (> tenfold), was found in 7 of the 10 annotation groups listed in Supplementary Table 3 (asterisks). *Ahsg* belongs to the cystatin superfamily of cysteine protease inhibitors, encoding for the negative acute phase glycoprotein fetuin-A, whose main function concerns mineralized matrix metabolism[25]. Despite its strong induction in hypoxic fetal kidneys, we did not detect a significant rise in serum fetuin-A levels in hypoxic fetuses (68.7 ± 4.3 µg/ml) compared to normoxic controls (70.5 ± 3.8 µg/ml) (Fig. 2d), nor in their mothers (Fig. 2e: normoxia: 190.0 ± 14.1 µg/ml vs. hypoxia: 172.1 ± 14.8 µg/ml). This is not surprising as under normal conditions the liver is the main source of circulating fetuin-A (600-fold higher than in the fetal kidney), whereas in our model, fetal hypoxia-induced local, "ectopic" fetuin-A expression in the kidney, at a level which is still 60-fold lower than in the liver (Fig. 2b: note that the expression levels of fetuin-A in kidney and liver are shown on a logarithmic scale). These findings provide evidence that the induction of

fetuin-A in hypoxic fetal kidneys does not have a systemic functional relevance, but rather a local, important protective role in the developing kidney.

**Fetuin-A is produced locally in the proximal tubulus under hypoxic conditions.** To further address the functional relevance of *Ahsg* expression in fetal hypoxic kidneys, we determined its precise localization. Whole-mount in situ hybridization disclosed fetuin-A mRNA synthesis in cortical regions of hypoxic fetal kidneys, but not in normoxic kidneys (Supplementary Fig. 4a, b). Immunofluorescent staining for fetuin-A protein pinpointed its location more precisely to the outer renal cortex, just below the nephrogenic zone (Supplementary Fig. 4c). No signal was detected in the inner cortex or medulla. Close up, Fig. 3 shows immunohistochemistry of fetal kidney proximal tubules (PT) (Fig. 3a, c, e, g) or tubule lumen (Fig. 3b, d, f, h). Strong fetuin-A staining demarcated PT cells regardless of the oxygen conditions (Fig. 3a, c). This is due to filtration and uptake of systemic fetuin-A into the PT via megalin-dependent endocytosis[26], masking any fetuin-A locally produced in the hypoxic fetal kidney. To selectively visualize fetuin-A protein of renal origin, we employed a genetic approach to block endocytosis into the PT, an alternative method to the pharmacological inhibition of megalin-dependent endocytosis using His-sRAP (histidine-tagged soluble receptor-associated protein)[26]. *Clcn5* knock-out (KO) mice[27] show severely impaired endocytosis of low molecular weight proteins in the PT, mimicking Dent's disease[28]. Normoxic *Clcn5* KO kidneys lacked the prominent fetuin-A staining in PT cells (Fig. 3e). Instead, a strong intraluminal signal was detected (Fig. 3f), which was not present in wild-type (wt) samples (Fig. 3b), highlighting the impaired endocytic phenotype of *Clcn5* KO mice. However, hypoxic *Clcn5* KO kidneys showed strong fetuin-A staining in the PT (Fig. 3g) in addition to the luminal signal (Fig. 3h), providing evidence that the observed cellular fetuin-A staining genuinely originated in the PT. Double immunofluorescence staining for fetuin-A and different renal segment markers (Fig. 3i–x)

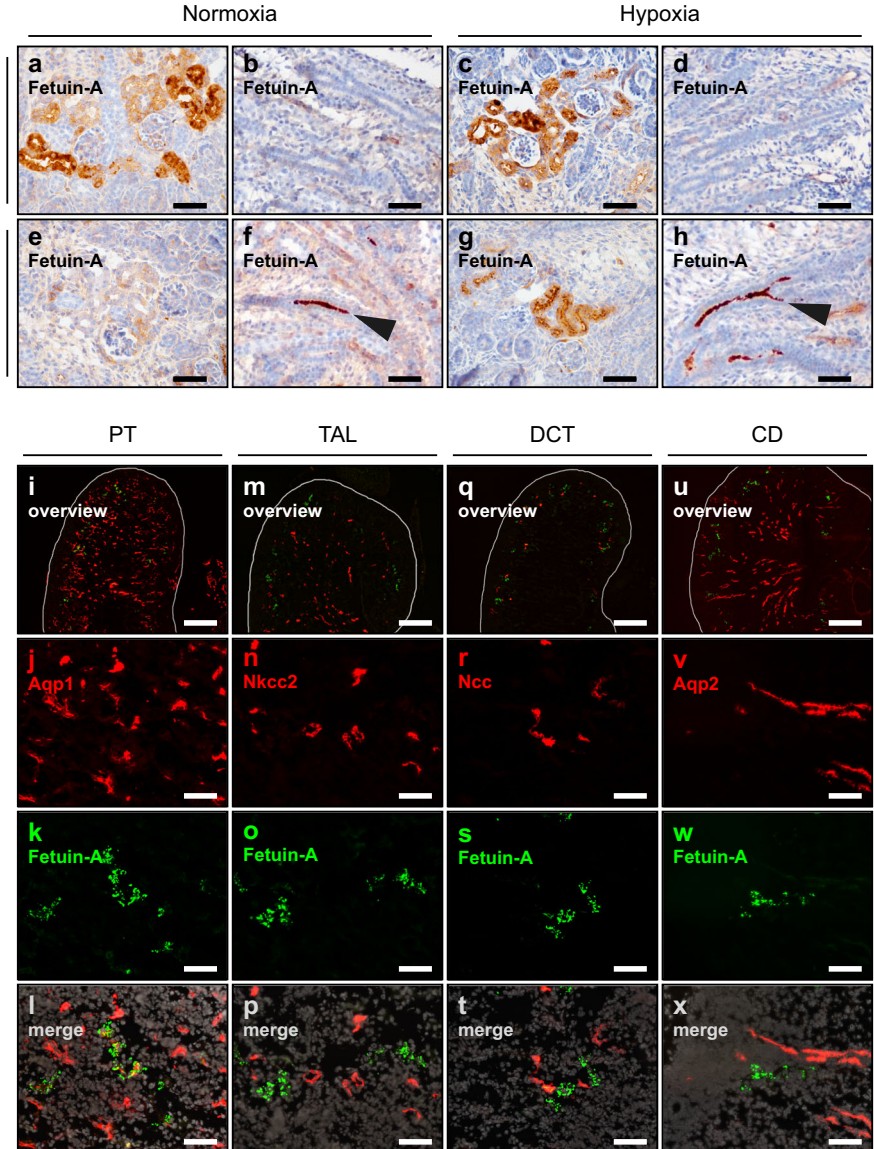

**Fig. 3 Fetal hypoxia induces fetuin-A expression in the proximal tubulus. a–h** Fetuin-A staining on E18.5 kidney sections of normoxic (**a**, **b**, **e**, **f**) or hypoxic (**c**, **d**, **g**, **h**), wild type (**a–d**) or *Clcn5* KO (**e–h**) fetuses. Arrowheads indicate intraluminal fetuin-A staining that results from the impaired endocytosis of low molecular weight proteins in the PT of *Clcn5* KO mice (**f**, **h**). **i–x** Immunofluorescence staining of the indicated nephron segment marker proteins (red) and fetuin-A (green) on E18.5 kidney sections. PT, proximal tubulus (**i–l**); TAL, thick ascending limb (**m–p**); DCT, distal convoluted tubulus (**q–t**); CD, collecting duct (**u–x**). Images are representative of at least three independent antibody stainings (**a–x**). Scale bar = 50 µm (**a–x**), except overview images for which the scale bar = 300 µm.

confirmed that fetuin-A was expressed only in the PT of hypoxic fetal kidneys.

**Ahsg harbors putative HIF-binding sites overlapping with enhancer regions.** Having shown that fetuin-A is locally produced in hypoxic fetal kidneys, we assessed whether the expression of *Ahsg* was directly activated by hypoxia. To check for potential HIF-binding sites (hypoxia response elements-HRE) in the human *AHSG* locus, we made use of HIF-1-alpha and HIF-2-alpha ChIP-seq data sets derived from hypoxic MCF7 cells[29]. We identified a cluster of potential HREs near exon 4 of human *AHSG* that overlapped with H3K27Ac and H3K4Me1 (chromatin marks of active enhancer elements[30]) and DNaseI hypersensitivity (Supplementary Fig. 5a). Another putative HRE was located in intron 1. Screening *Ahsg* genes of 15 species for the presence of the consensus HIF-binding sites (RCGTG) 10 kb up- and

downstream of the ATG revealed a peak 1–5 kb downstream of the ATG with an average number of 2 HREs per 1 kb window (Supplementary Fig. 5b). Notably, not only the annotated human ChIP-seq HIF sites localized within this peak, but also four potential mouse HREs. Alignment of the latter with enhancer marks revealed a close association with H3K27Ac, H3K4Me1 and DNaseI hypersensitivity (Supplementary Fig. 5c). A complete list of the identified sites can be found in Supplementary Table 5.

**Hypoxia activates fetuin-A transcription in vitro and promotes the expression of fibrotic marker genes in fetal organs.** Five putative HREs of mouse *Ahsg* and their surrounding DNA, alongside with nonsense mutations of these sites, were cloned into luciferase reporter plasmids (Fig. 4a; DNA sequences in Supplementary Table 6). Normal rat kidney epithelial (NRK) cells transfected with reporter plasmids containing only the putative

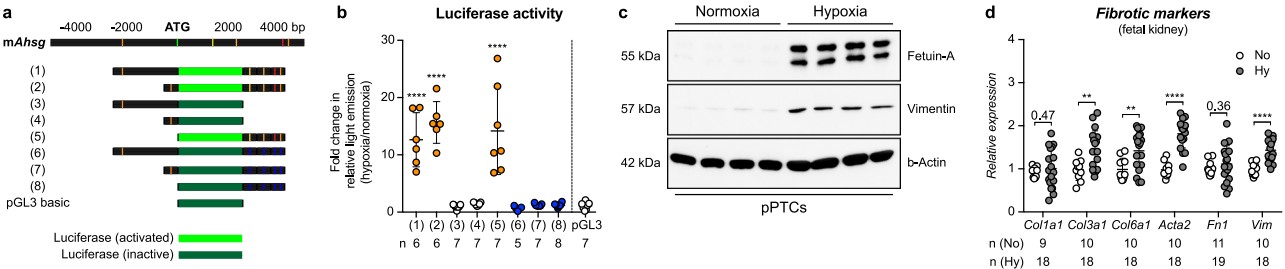

**Fig. 4 Hypoxia activates fetuin-A expression in vitro. a** Depiction of the potential HREs of mouse *Ahsg* that were used to generate the luciferase reporter gene constructs (1) to (8). Mutated HREs are shown in blue. **b** Mean ± SEM of luciferase activity in NRK cells individually transfected with the reporter constructs depicted in **a**, showing the fold-change in light emission between hypoxic and normoxic culture conditions. Each transfection condition is compared to the empty vector control (pGL3). N = independent experiments. Ordinary one-way ANOVA with Dunnett's multiple comparisons test. **c** Expression of fetuin-A and vimentin in primary mouse proximal tubular cells (pPTCs) isolated from four different mice cultured under normoxic or hypoxic conditions. Images are representative of two independent Western blots. Uncropped blots in Source Data. **d** Relative mRNA expression levels of collagens (*Col1a1, Col3a1,* and *Col6a1*), α-smooth muscle actin (*Acta2*), fibronectin (*Fn1*), and vimentin (*Vim*) in kidneys from normoxic (white circles) or hypoxic fetuses (gray circles). Data is shown as mean ± SEM. N = fetal kidneys. Unpaired 2-sided *t*-test (with Welch's correction for *Col1a1* and *Fn1*). Individual *P*-values are denoted above the comparison lines. (****$P < 0.0001$; **$P < 0.01$). Source data are provided as a Source Data file.

−2 kb HRE did not show increased luciferase activity under hypoxic conditions (Fig. 4b). Conversely, NRK cells carrying reporter plasmids containing the downstream HREs significantly increased luminescence in hypoxia. No increased luciferase activity was detected when these HREs were mutated (Fig. 4b). Furthermore, there was no enhanced luciferase activity when up- and downstream HREs were combined. Thus, only the HREs located downstream of the ATG conferred hypoxia inducibility to the mouse *Ahsg* gene. Besides the activation of luciferase from reporter constructs, hypoxia also triggered the production of fetuin-A protein in primary mouse PT cells (pPTCs), NRK cells and in the human kidney cell line HK-2 (Fig. 4c and Supplementary Fig. 6a, b, respectively). Taken together, these findings identified fetuin-A as an evolutionary conserved HIF-dependent target gene. Moreover, hypoxia not only promoted the expression of fetuin-A, but also stimulated the expression of several fibrotic marker genes in pPTCs (Fig. 4c), fetal kidneys (Fig. 4d), and fetal lungs and hearts (Supplementary Fig. 6c, d). However, whereas as the epithelial organs (lung and kidney) showed a broad activation of fibrotic genes, the response in the heart was more blunted and did not include an enhanced transcription of collagens.

**Fetuin-A deficiency aggravates CKD progression in hypoxic IUGR kidneys.** To investigate how the induction of fetuin-A in fetal hypoxic IUGR kidneys affects renal function in the long-term, we measured urinary protein levels and determined the glomerular filtration rate (GFR) in adult wild-type (wt) and fetuin-A (*Ahsg*) KO mice[31] (Fig. 5a, b). GFR was reduced, whereas proteinuria was severely increased in both sexes of 9-week-old *Ahsg* KO animals and in mice exposed to fetal hypoxia compared to normoxic controls. While the GFR reduction was comparable between the sexes, the degree of proteinuria was much higher in males than in females. Furthermore, assessment of fibrotic tissue remodeling revealed enhanced expression of collagens, showing the highest expression levels in kidneys of hypoxic *Ahsg* KO mice (Fig. 5c–e), which in histological examinations showed a broader staining pattern with multiple collagen bundles extending deeper into subcortical regions (Fig. 5f–i). Besides these genes, we also found non-collagenous fibrosis markers to be significantly induced in adult hypoxic KO kidneys (Supplementary Fig. 7), but also in *Ahsg* KO-derived pPTCs cultured under hypoxic conditions (Figs. 4c and 5j). Importantly, these results clearly show that renal function was most affected in *Ahsg* KO mice, showing an additive effect of hypoxia and fetuin-A deficiency.

**Fetuin-A attenuates the hypoxia-induced expression of fibrosis markers by antagonizing TGF-β signaling.** The additional increase in the expression levels of fibrotic markers in the kidneys of hypoxic *Ahsg* KO mice compared to hypoxia alone (Fig. 5 and Supplementary Fig. 7) occurred despite similar mRNA levels in these two groups of transforming growth factor beta-1 (*Tgfb1*), a potent inducer of fibrosis (Fig. 6f). We elucidated this finding in vitro, using freshly isolated pPTCs, and found that the supplementation of fetuin-A to the culture medium blunted the hypoxia-induced increased gene expression levels of fibrotic markers (downward pointing triangles in Fig. 6a–d and in Supplementary Fig. 8a–f). Moreover, this diminishing effect was also observed at the protein level (Fig. 6e). However, when a similar amount of BSA was applied instead of fetuin-A, the expression of fibrotic markers was not reduced (upward pointing triangles in Fig. 6c, d and in Supplementary Fig. 8d–f). Stimulation of pPTCs with recombinant TGF-β1 resulted in robust phosphorylation of its intracellular signal transducer Smad3 (Fig. 6g and Supplementary Fig. 8g). This activation was more than three times stronger in *Ahsg* KO pPTCs compared to wt cells (Fig. 6h), showing that *Ahsg* KO cells responded more vividly to TGF-β1. In contrast, adding fetuin-A before TGF-β1 treatment decreased Smad3 phosphorylation (Supplementary Fig. 8g). These findings are in line with previous reports, describing fetuin-A as a soluble decoy receptor protein mimicking TGF-β type II receptor and cytokine antagonist[32,33]. Collectively, our results suggest that fetuin-A reduces hypoxia-induced renal fibrosis by direct antagonization of TGF-β1 signaling.

**Renal infiltration and polarization of pro-inflammatory M1 macrophages during fetal hypoxia is mitigated by fetuin-A.** Another important cell type that contributes to the progression of renal injury are macrophages[34,35]. During its development, the kidney is first populated by embryo-derived, long-lived, self-renewing F4/80$^{hi}$CD11b$^{low}$ cells, which maintain a resident population of macrophages[36]. This group is complemented by bone-marrow derived circulatory F4/80$^{low}$CD11b$^{hi}$ macrophages, which infiltrate and patrol, but rarely colonize the kidney except during renal injury[37]. Besides this classification, macrophages can be further categorized according to their polarization into pro-inflammatory M1 or anti-inflammatory M2 cells. We used a FACS approach to characterize the macrophage populations in E18.5 hypoxic or normoxic fetal kidneys. Strikingly, in hypoxia the composition of renal macrophages was shifted towards infiltrating F4/80$^{low}$CD11b$^{hi}$ cells, partially replacing the resident

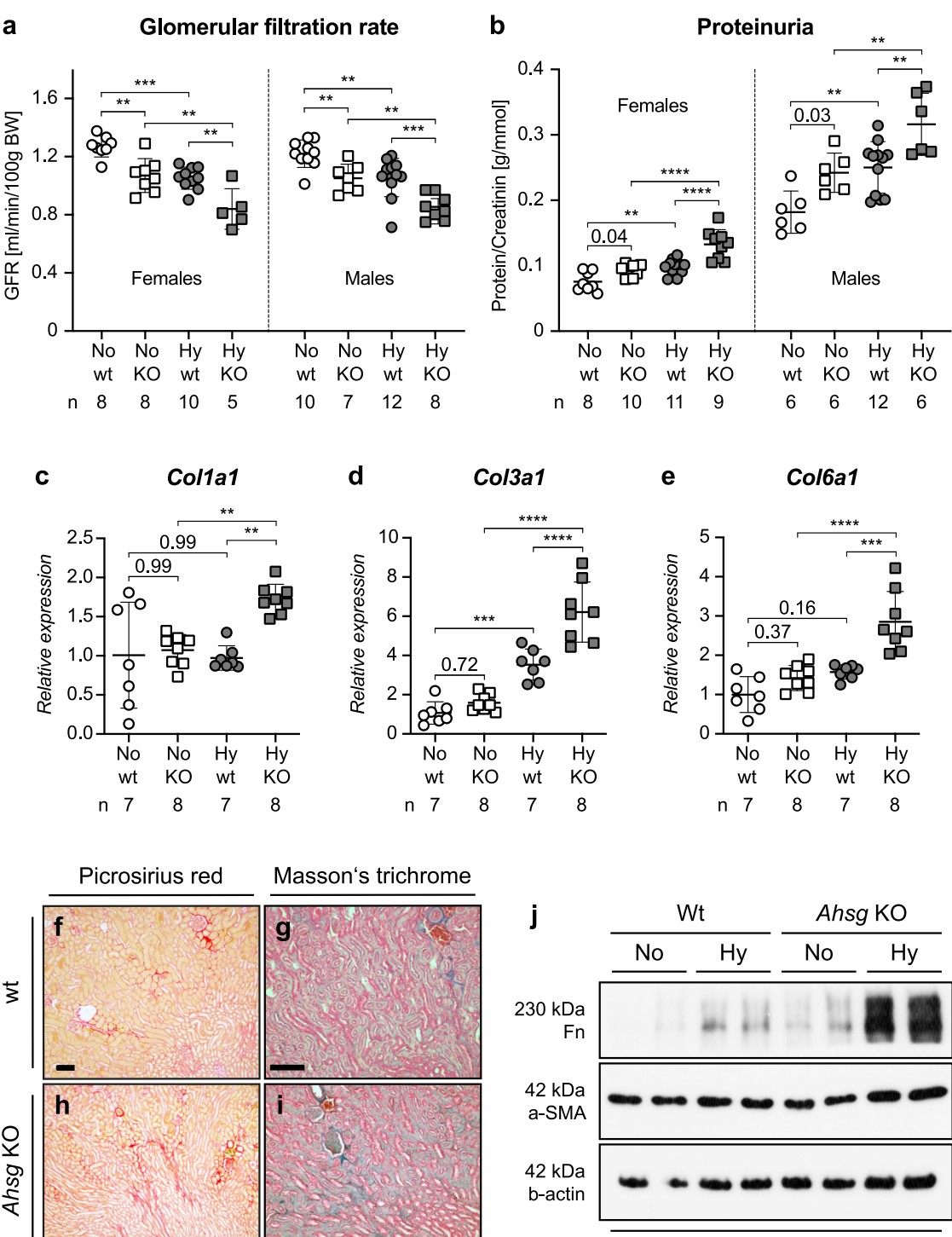

**Fig. 5 Fetuin-A deficiency aggravates CKD progression in hypoxic IUGR kidneys. a**, **b** Decline of renal function in adult hypoxic offspring showed additive effects of hypoxia and fetuin-A deficiency. The decline in GFR was indistinguishable between the sexes with the greatest functional reduction in hypoxic *Ahsg* KO animals (**a**). The incline in proteinuria (protein/creatinine ratio) was more pronounced in males than in females. For both sexes, hypoxic *Ahsg* KO animals had the highest ratios (**b**). Male and female samples are analyzed separately. **c**–**e** Relative mRNA expression levels of *Col1a1* (**c**), *Col3a1* (**d**), and *Col6a1* (**e**) were markedly enhanced in kidneys of hypoxic *Ahsg* KO offspring. **f**–**i** Histological depiction of collagen using picrosirius red (**f**, **h**) or Masson's trichrome staining (**g**, **i**) showed a stronger, more intricate pattern on kidney sections of hypoxic *Ahsg* KO offprings compared to controls. Images are representative of at least three independent experiments. Scale bar = 100 μm. **j** Primary proximal tubular cells (pPTCs) isolated from two different wt or *Ahsg* KO mice exhibit enhanced expression of fibronectin and α-smooth muscle actin (α-SMA) protein upon culture in hypoxic conditions. Images are representative of three independent Western blots. Uncropped blots in Source Data. Data were analyzed from N = hypoxic or normoxic offspring and are presented as mean ± SEM (**a**–**e**). Ordinary one-way ANOVA with Dunnett's multiple comparisons test (**a**–**b**) or Ordinary one-way ANOVA with Tukey's multiple comparisons test (**c**–**e**). Individual *P*-values are denoted above the comparison lines (**b**, **c**, **e**). (****P < 0.0001; ***P < 0.001; **P < 0.01). Source data are provided as a Source Data file.

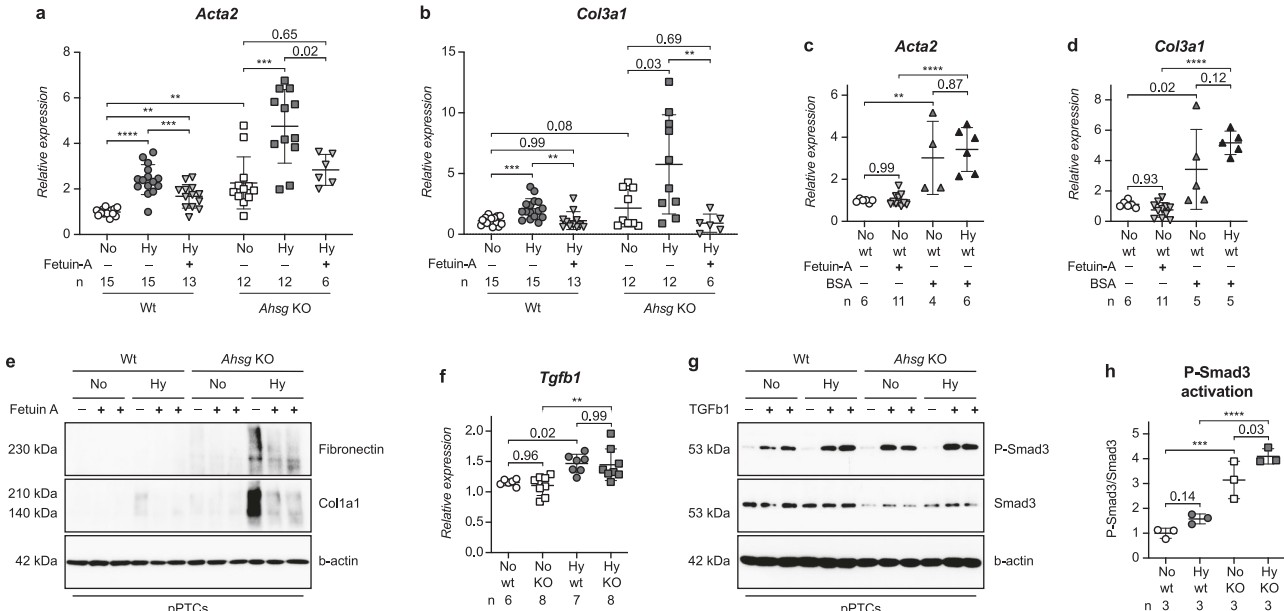

**Fig. 6 Fetuin-A attenuates hypoxia-induced expression of fibrotic markers. a, b** Fetuin-A supplementation (downward pointing triangles) attenuated the hypoxia-induced expression of the fibrotic markers *Acta2* (**a**) or *Col3a1* (**b**) in pPTCs. Wt and *Ahsg* KO samples are analyzed separately. Unpaired two-tailed *t*-test with Welch's correction (only for comparison of normoxic wt and normoxic *Ahsg* KO samples). **c, d** BSA supplementation (upward pointing triangles), did not reduce the expression of *Acta2* (**c**) or *Col3a1* (**d**) in pPTCs. **e** Fetuin-A supplementation also reduced the expression of fibronectin and collagen type I (Col1a1) protein in pPTCs. Images are representative of two Western blots. Uncropped blots in Source Data. **f** Relative mRNA expression levels of *Tgfb1* shown as mean ± SEM were markedly enhanced in kidneys of hypoxic offspring, regardless of genotype. N = hypoxic or normoxic offspring. **g** TGF-β1 treatment and hypoxia had an additive effect on the phosphorylation of Smad3 in pPTCs. Images are representative of three Western blots. Uncropped blots in Source Data. **h** Quantification of Smad3 activation shown in **g**. N = three independent Western blots. Ordinary one-way ANOVA (Fisher's LSD test). Data were analyzed from N = pPTCs derived from kidneys of wt or *Ahsg* KO mice and are presented as mean ± SEM (**a–d**). Ordinary one-way ANOVA with Tukey's multiple comparisons test (**a–d, f**). Individual P-values are denoted above the comparison lines (**a–d, f, h**). (****P < 0.0001; ***P < 0.001; **P < 0.01). Source data are provided as a Source Data file.

population (Fig. 7a–f). Furthermore, whereas most macrophages were M2 polarized (CD206+) in normoxic kidneys, the majority of macrophages isolated from hypoxic kidneys had adapted a M1, pro-inflammatory phenotype (Fig. 7g–k and Supplementary Fig. 9). Importantly, the fraction of M1 polarized macrophages was even more prominent in hypoxic *Ahsg* KO samples (Table 1). Taken together, we show that hypoxia promotes the infiltration and polarization of pro-inflammatory M1 macrophages (CD206−) in the kidney, suggesting that fetuin-A was associated with an overall anti-inflammatory milieu.

**Calcium mineral particles accumulate in hypoxic IUGR kidneys of *Ahsg* KO mice.** One cause for the enhanced inflammatory phenotype in hypoxic *Ahsg* KO kidneys could be renal calcification, since fetuin-A KO mice are prone to soft tissue calcification[38]. However, we did not detect overt calcification in the fetal kidneys of hypoxic *Ahsg* KO mice with classic methods such as von-Kossa staining. We therefore applied a more sensitive method to test whether the expression of fetuin-A in fetal hypoxic kidneys affected mineralized matrix handling, and probed for the presence of calcium containing microparticles by incubating freshly cut kidney sections of E18.5 embryos with ATTO 488 fluorescently labeled fetuin-A (488-FA)[39,40]. Owing to the high-affinity binding of fetuin-A to calcium phosphate[41], 488-FA staining is more sensitive to detect calcium containing matrix and cell remnants than the commonly used mineral staining protocols. Thus, positive 488-FA staining in the absence of von-Kossa or Alizarin-Red staining also highlights structures merely enriched with calcium, including amorphous calcium-phosphate aggregates that often precede overt calcifications[42]. 488-FA staining revealed in normoxic wt kidneys intense labeling of the

PT, a site of major calcium resorption[43] and thus also of mineralized matrix handling (Fig. 8a). PT staining intensity was reduced in hypoxic wt and increased in hypoxic *Ahsg* KO (Fig. 8b, c). Only hypoxic *Ahsg* KO kidneys also showed a granular staining pattern in the papillary region and less frequently in the nephrogenic region of the outer cortex (arrowheads in Fig. 8f, i), indicating bulk mineral deposition in the absence of endogenous fetuin-A. Excess bulk mineral or cellular debris is often found at sites of enhanced cell death[44]. Indeed, TUNEL staining confirmed apoptosis in hypoxic *Ahsg* KO kidneys, but not in hypoxic wt kidneys or normoxia (Supplementary Fig. 10a–c). Cleaved caspase-3 immunostaining in hypoxic *Ahsg* KO kidneys further corroborated cell death in these kidneys (Supplementary Fig. 10d–f). Collectively, these data illustrate the role of fetuin-A in binding and clearance of mineralized matrix in the kidney.

**Fetuin-A supplementation reduces the expression of fibrotic markers upon ischemia-reperfusion injury.** In the previous sections we have presented results that describe fetuin-A as an important player in Barker's hypothesis, counteracting multiple disadvantageous processes in the fetal kidney. Because of the underlying pathophysiological mechanisms we anticipated that the protective role of fetuin-A is not restricted to the fetus, but can be extended to offset similar harmful processes in hypoxia-related injury in adult animals. Thus, in a final step, we performed an interventional study using a mouse model of ischemia-reperfusion injury (IRI)[45]. Here, renal blood flow is transiently stopped to induce hypoxic damage in the kidney (ischemia), which is further exacerbated upon the restoration of renal circulation (reperfusion). Similar to our fetal model, we showed that tissue damage in IRI kidneys was associated with the deposition

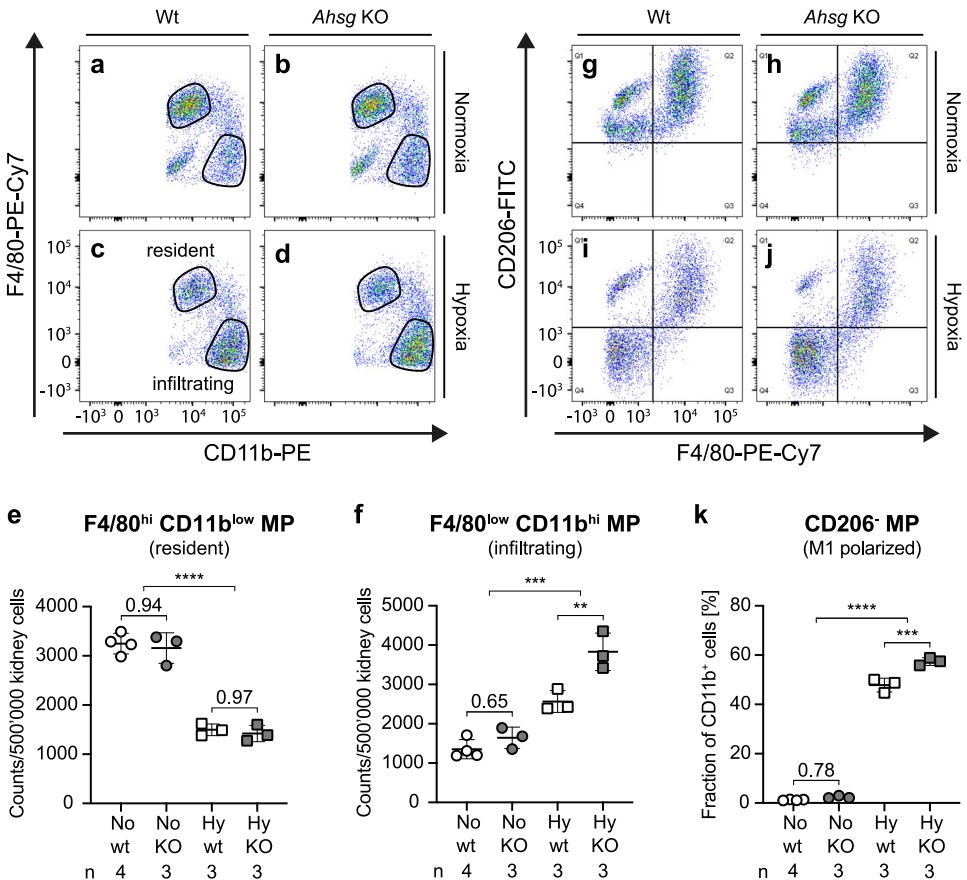

**Fig. 7 Fetuin-A mitigates infiltration and polarization of pro-inflammatory M1 macrophages. a–d** Under normoxic conditions (**a**, **b**), the majority of renal macrophages exhibits a F4/80$^{hi}$CD11b$^{low}$ phenotype, indicating resident macrophages. The number of these cells was generally reduced under hypoxic conditions. Under hypoxic conditions (**c**, **d**), the cell count of infiltrating macrophages (F4/80$^{low}$Cd11b$^{hi}$) was increased. Lack of fetuin-A even more stimulated the infiltration of macrophages into hypoxic fetal kidneys (**d**). Images are representative of 3 (**b–d**) or 4 (**a**) sorted kidneys. **e** Quantification of resident macrophages shown in **a–d**. Unpaired two-tailed *t*-test (for comparison of normoxic and hypoxic condition). **f** Quantification of infiltrating macrophages shown in **a–d**. Unpaired two-tailed *t*-test with Welch's correction (for comparison of normoxic and hypoxic condition). **g–j** Under normoxic conditions (**g**, **h**), the majority of renal macrophages exhibits a M2 CD206$^+$ anti-inflammatory phenotype (depicted in the upper two quadrants). Hypoxic conditions (**i**, **j**) promoted the polarization of M1 CD206$^−$ pro-inflammatory macrophages (depicted in the lower two quadrants). This polarization is even more pronounced in fetal kidneys of *Ahsg* KO mice (**j**). Images are representative of three (**h–j**) or four (**g**) sorted kidneys. **k** Quantification of the lower two quadrants (CD206$^−$ macrophages) of the FACS blots shown in **g–j**. Unpaired two-tailed *t*-test with Welch's correction (for comparison of normoxic and hypoxic condition). Data were analyzed from $N$ = fetal kidneys and are presented as mean ± SEM (**e**, **f**, **k**). Ordinary one-way ANOVA with Tukey's multiple comparisons test (**e**, **f**, **k**). Individual *P*-values are denoted above the comparison lines (**e**, **f**, **k**). (****$P$ < 0.0001; ***$P$ < 0.001; **$P$ < 0.01). Source data are provided as a Source Data file.

**Table 1 Number of macrophages expressing markers of M1 polarization per 500,000 cells.**

| Markers | Normoxia | | | Hypoxia | | |
|---|---|---|---|---|---|---|
| | **wt** | **KO** | ***P*-Value** | **wt** | **KO** | ***P*-Value** |
| CD11c$^+$ | 1256 ± 214 | 1589 ± 201 | 0.138 | 2209 ± 325 | 2988 ± 208 | 0.046 |
| CD68$^+$ | 48 ± 11 | 58 ± 4 | 0.245 | 121 ± 21 | 167 ± 9 | 0.047 |
| CD80$^+$ | 265 ± 61 | 357 ± 51 | 0.132 | 442 ± 42 | 674 ± 31 | 0.003 |
| CD86$^+$ | 796 ± 113 | 1037 ± 93 | 0.052 | 2044 ± 387 | 2503 ± 219 | 0.218 |

Data were analyzed from $N$ = 3 or 4 fetal kidneys and are presented as mean ± SEM. Unpaired two-tailed *t*-test.

of calcium containing microparticles (Fig. 8k), which were not found in controls (Fig. 8j). The presence of these deposits in IRI kidneys not only validated the superiority of the 488-FA staining approach to detect early calcium biominerals, but also corroborated our findings in fetal hypoxia, revealing an imbalance between calcium mineral release and clearance upon hypoxic tissue damage. Furthermore, daily administration of fetuin-A for

4 days, starting immediately after IRI surgery, resulted in a marked decrease of *Col1a1* and *Col3a1* expression compared to mice treated with physiological saline solution (Fig. 8l, m). In this regard, it was reported in rats that peripheral fetuin-A administration could prevent excessive cerebral ischemic tissue injury[46]. Our results provide strong evidence that fetuin-A supplementation at the time of injury (e.g., ischemia-reperfusion injury in

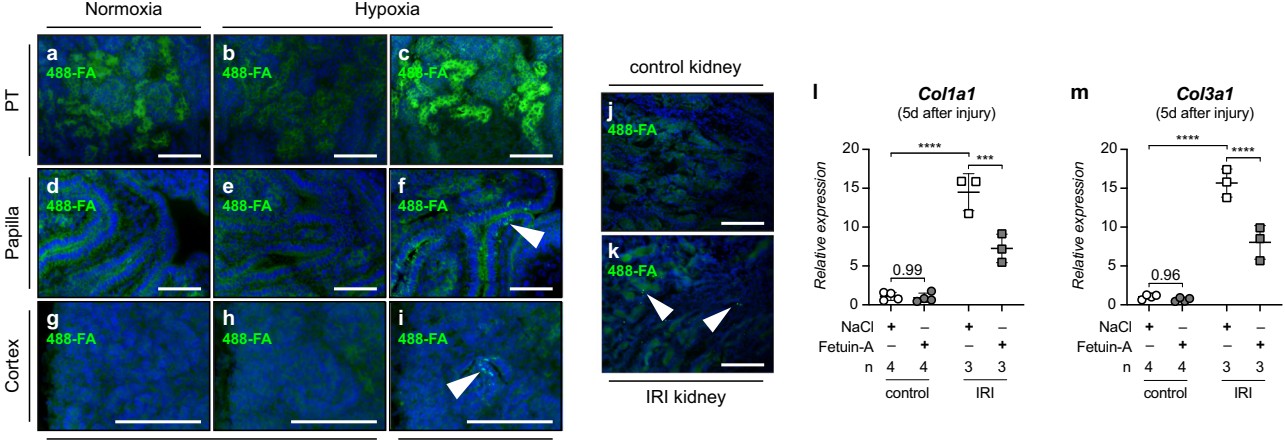

**Fig. 8 Fetuin-A supplementation reduces the expression of fibrotic markers upon hypoxia-related injury. a–i** Fetuin-A deficiency promotes accumulation of calcium mineral particles in hypoxic fetal kidneys. Calcium biominerals were detected by ATTO 488 fluorescently labeled fetuin-A (488-FA). Compared to normoxic or hypoxic wt mice, hypoxic *Ahsg* KO mice exhibited the strongest 488-FA staining intensity in the proximal tubulus (PT), indicative of an increased mineralized matrix turnover (**a–c**). Arrowheads in **f** and **i** point towards granular staining pattern in the papilla and cortex, respectively, reflecting bulk accumulation of 488-FA in kidneys of hypoxic *Ahsg* KOs. Such granules were not detectable in wt samples (**d, e, g, h**). **j, k** Ischemia-reperfusion injury (IRI) induces calcium mineral particles in adult kidneys. A granular staining pattern indicative of bulk accumulation of 488-FA at sites of calcium deposits was only present in IRI kidneys (**k**), but not in controls (**j**). **l, m** Fetuin-A supplementation reduced the expression of the fibrotic markers *Col1a1* (**l**) and *Col3a1* (**m**) in IRI kidneys 5 days after injury. No effect is seen in mice treated with physiological saline solution (NaCl). Data were analyzed from $N =$ kidneys and are presented as mean ± SEM. Ordinary one-way ANOVA with Dunnett's multiple comparisons test. Individual *P*-values are denoted above the comparison lines. (****$P < 0.0001$; ***$P < 0.001$). Images are representative of at least three independent experiments (**a–k**). Scale bar = 100 µm (**a–k**). Source data are provided as a Source Data file.

diverse organ systems) could be a promising therapeutic approach against hypoxia-induced mineral stress and fibrotic tissue remodeling, particularly in conditions associated with fetuin-A depletion, such as CKD or acute inflammation.

## Discussion

In the present study, we employed a comprehensive array of in vitro, in vivo and in silico methods to characterize IUGR secondary to chronic fetal hypoxia. Using the kidney as readout, we determined the short and long-term consequences on renal development and function, and elucidated molecular mechanisms linking fetal hypoxia and progression to renal fibrosis in adults. Importantly, we identified fetuin-A (*Ahsg*) as an evolutionary conserved HIF target gene, protecting the kidney from hypoxia-induced renal damage by counteracting not only cellular debris-mediated mineralization, but also macrophage polarization and fibrotic remodeling.

Several groups including ours have established hypoxic IUGR models in rodents either by partial ligation of the uterine artery[47,48], or by exposure to a low-oxygen atmosphere[49–51]. Yet, although all studies found nephron numbers reduced by 25–30%, the reported adult phenotypes differed, comprising reduced GFR[47], aberrant expression of angiotensin II receptors, or urine concentration defects[47,51]. The variability is most likely caused by the specific hypoxia protocols employed, especially differences in exposure time, developmental stage at start of exposure, or the severity of hypoxia. Mechanistically, two studies proposed that fetal hypoxia altered canonical Wnt signaling, thus impairing nephrogenesis[49,51]. Besides the kidney, late gestational hypoxia in combination with high-salt diet was also shown to promote arterial stiffness due to increased deposition of collagen in the vessel walls of mice[52]. In contrast to some of the models mentioned above, which in some instances needed a second hit to promote a phenotype, we established and validated a robust murine model of chronic fetal hypoxia-induced IUGR, where late

gestational hypoxia alone was sufficient to reduce GFR and to induce proteinuria in adult IUGR mice. Under these circumstances, lack of fetuin-A further aggravated the chronic damage due to fetal hypoxia. If these findings were to be translated into a clinical perspective, then hypoxic or *Ahsg* KO animals would have a moderate to high risk for CKD (Supplementary Fig. 11), thus substantiating and giving a mechanistic insight into Barker's hypothesis.

In our microarray, *Ahsg* showed the highest induction and we further revealed that the most relevant HIF-binding sites are located downstream of the ATG. This is in line with a previous genome-wide mapping of HIF-binding sites, describing a broad peak of enrichment downstream of the transcriptional start site[53]. Taken together with this finding, our results provide strong evidence that *Ahsg* is a HIF target gene.

Hypoxia-induced expression of fetuin-A might represent a general reactive mechanism of extra-hepatic tissues involved in mass transport of solutes across an epithelium to safeguard the proper handling of calcium and phosphate locally, at sites of increased mineral stress[54]. It would be interesting to assess other non-renal epithelia (e.g., choroid plexus, intestine, salivary glands) for their response to hypoxia. The liver on the other hand, might not equally respond to reduced oxygen levels, since constitutive hepatic fetuin-A synthesis is regulated by a strong promoter driving high systemic levels[55]. A reason for the specific induction of fetuin-A expression in fetal hypoxic kidneys might be the fetal circulation with its extra- and intracardiac shunt mechanisms, delivering the most highly oxygenated blood to the brain and myocardium. This further reduces the oxygenation of more peripheral organs including the developing kidney already at normoxic conditions. Thus, in the hypoxic group, the fetal kidney is exposed to much more stringent hypoxic conditions, which then responds to this severe hypoxic environment with the induction of fetuin-A. Unlike mRNA, exact quantification of fetuin-A protein produced locally in the kidney is not trivial due to the uptake of filtered hepatic fetuin-A into the cells of the

proximal tubules, which cannot be easily distinguished from fetuin-A of renal origin. However, given the fourfold size difference between liver and kidneys and the roles that these two organs play in fetuin-A distribution (the liver is releasing fetuin-A into the circulation for systemic use, whereas the fetuin-A produced in the kidneys upon hypoxic injury plays a local role with no systemic relevance), we believe that the concentration of locally produced fetuin-A in the kidneys is sufficiently high to produce a significant renal protection.

488-FA staining revealed the presence of bulk mineral particles in the kidney of hypoxic fetuses. These granules most likely represent deposits of mineralized debris commonly detected by electron microscopy at sites of excessive tissue damage or remodeling[42]. In this regard, it has been suggested that fetuin-A plays a role in tissue remodeling during embryogenesis, its expression peaking during the transition from organogenesis and histogenesis, when cells acquire their final phenotype, but is lost thereafter[56,57]. Yet, recent evidence shows that fetuin-A expression is reactivated upon ischemic brain damage, aiding tissue repair[58]. In the developing kidney, prenatal hypoxia induces apoptosis[59,60] and was reported to cause renal vascular dysfunction[61,62], giving rise to excess amounts of cellular debris that could serve as mineralization foci. Damaged or dying cells suffer calcium overload, which is characterized by calcium accumulation in mitochondria or apoptotic bodies, respectively[63–65]. Concomitant low levels of ATP (malfunctioning mitochondria) and pyrophosphate levels (an important inhibitor of calcification[66]) increases the calcification propensity of these organelles. The formation and local deposition of solid mineral from saturated mineral solutions is regulated by mineral binding proteins like fetuin-A, which stabilize mineral as colloidal complexes and mediate their clearance and recycling[67]. Thus, a lack of fetuin-A increases the risk of mineral debris deposition and calcification. In addition, fetuin-A was also shown to inhibit apoptosis and to augment phagocytosis of apoptotic cells[68,69], reducing the amount of mineral debris generated and enhancing its removal, respectively. Uptake of fetuin-A containing calci-protein particles (CPPs) from the circulation depends on their maturation state[39]. Whereas early amorphous CPPs are preferentially cleared by endothelial cells, crystalline CPPs are taken up predominantly by macrophages through a scavenger receptor-A (SR-A) mediated pathway[70]. Interestingly, apart from fetuin-A, also about one third of the proteins constituting CPPs[23,24] were induced in our fetal hypoxic kidneys, including Apo-A1, Apo-A2 and transferrin. The latter was also shown to promote the expression of multiple pro-inflammatory chemokines in human proximal tubular epithelial cells[71]. Thus, hypoxic fetal kidneys seem to employ a mechanism that enhances the stabilization and clearance of mineral debris.

Chronic hypoxia, inflammation, and fibrotic tissue remodeling are tightly interwoven processes. Cells of the PT are especially vulnerable to stress due to their immense energy consumption required for the reabsorption of filtered molecules and ions. Thus, it is not surprising that these cells are the most preferred site of crystallization in the kidney[72] and that they respond with the release of cytokines (e.g., MCP1, TNFα, or TGF-β) during prolonged episodes of cellular stress[73,74]. Our GO biological pathways suggested the presence of immune cells, known to be universally recruited in tissue damage response and repair[75]. We further show that hypoxia induced a shift from anti-inflammatory M2 to pro-inflammatory M1 macrophages, which was even more prominent in fetuin-A KO mice. The polarization from M2 to M1 could be directly mediated by the low availability of oxygen during fetal hypoxia, since M1 macrophages rely on glycolysis to obtain energy, whereas M2 macrophages make use of oxidative metabolic processes[76]. The reduction in M2 macrophages might

also directly impair nephrogenesis in hypoxia, given the trophic function of tissue resident F4/80hiCD206+ M2 macrophages in the developing kidney[77,78]. These findings are in line with previous studies, reporting that fetuin-A dampens the pro-inflammatory phenotype of macrophages[54,79]. Here, the NLRP3 inflammasome is increasingly recognized to promote renal inflammation and fibrosis, contributing to the progression to CKD through enhanced secretion of the pro-inflammatory cytokines IL-1β and IL-18[80,81]. Although its main molecular components (NLRP3, ASC, and Caspase-1) are also expressed in renal tubular epithelial cells, macrophages are the main cell type sustaining an inflammatory reaction towards a multitude of endogenous, cell damage-associated molecular patterns, including basic calcium-phosphate crystals[82,83]. Excessive phagocytosis of these biominerals leads to calcium overload, lysosomal damage and the release of lysosomal enzymes, which sustains a vicious auto-amplification loop of necroinflammation[72]. In epithelial cells, NLRP3 was also described to augment TGF-β signaling independently of its function in inflammasomes[84]. These findings are in agreement with our observations showing that only kidneys from hypoxic *Ahsg* KO fetuses exhibited cell death and enhanced polarization of M1 macrophages, and that fetuin-A supplementation diminished the expression of fibrotic markers in pPTCs as well as upon IRI.

A previous study by Chatterjee and colleagues[85] has shown that fetuin-A promoted the polarization of M1 pro-inflammatory macrophages in adipose tissue, whereas we report here that fetuin-A reduces macrophage infiltration and M2 to M1 polarization. Our results do not invalidate in any way these previous findings, but on the contrary, strengthen the hypothesis that fetuin-A plays an important role in modulating macrophage responses. Both observations reflect the two sides of the same coin, the already known Janus nature of this protein, highlighting similarities: the ectopic (non-hepatic) expression of fetuin-A, but also disparities, namely different subsequent signaling events and phenotypic outcomes. On the one hand, excess lipids in adipose tissue due to obesity stimulate the local production of fetuin-A via the TLR4 and Nf-κB signaling cascade, creating a local microenvironment that stimulates M1 polarization and the release of pro-inflammatory cytokines along the same signaling axis[85–87]. Yet, such interactions have only been reported to take place in the adipose tissue[88]. On the other hand, in our case ectopic fetuin-A expression is downstream of HIF signaling triggered by severe hypoxia, for which the kidney is especially vulnerable. The downstream role of fetuin-A here is also very different: by binding excess calcium that is released from damaged or apoptotic cells and the formation of CPPs, fetuin-A reduces mineral stress in macrophages, thus protecting against some of the pro-inflammatory and harmful effects that are emanating from calcium-phosphate nanocrystals[54]. This function of fetuin-A does not seem to be mediated via TLR4, but involves the SR-A system[54,70].

In conclusion, we identified *Ahsg* as a hypoxia target gene that locally protects IUGR kidneys from chronic, progressive renal damage induced by prenatal hypoxia, and furthermore demonstrated the therapeutic potential of fetuin-A supplementation in acute ischemia-reperfusion injury. In Fig. 9, we propose a model in which the systemic function of liver-derived fetuin-A (green circles) can be locally enhanced upon hypoxic cellular stress. This locally produced, renal fetuin-A (yellow circles) provides a boost mechanism that augments the capacity in the kidney to clear the increased release of calcium minerals from stressed cells and to suppress inflammation. This in turn protects the kidney from further mineral stress by keeping the renal damage in check. An inability to activate the local fetuin-A response leads to an enhanced infiltration and polarization of pro-inflammatory M1

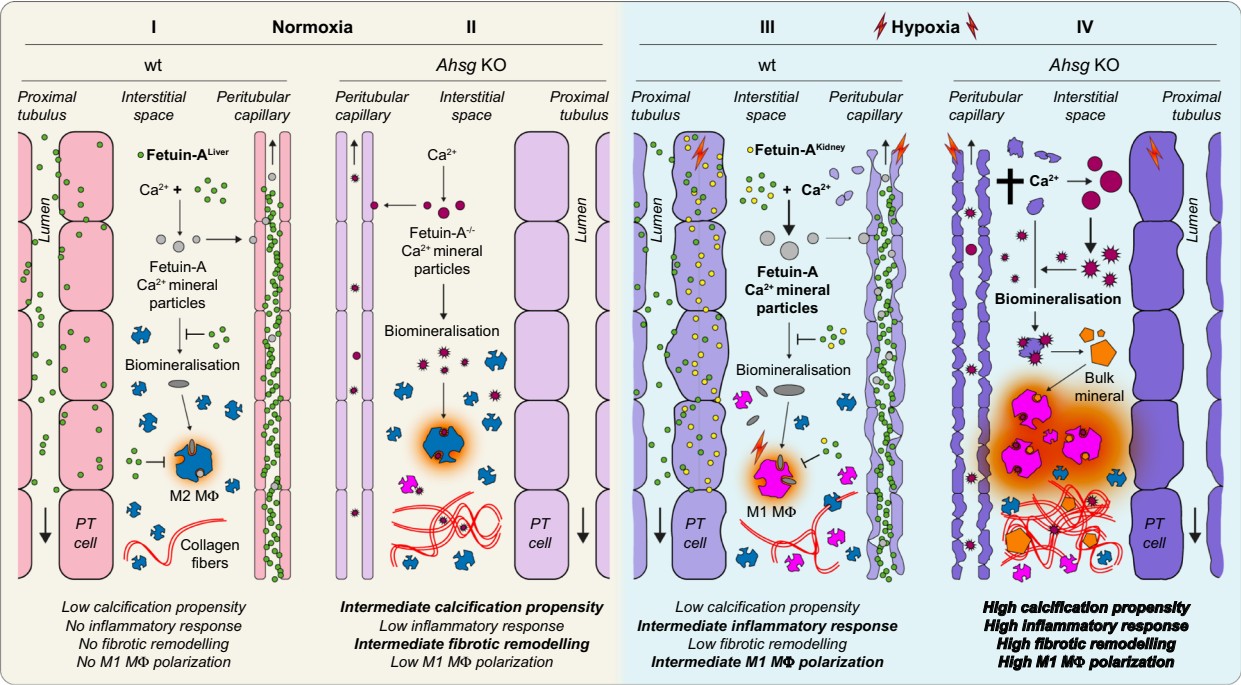

**Fig. 9 Proposed model.** Fetuin-A (*Ahsg*) is a HIF target gene that locally protects the kidney from hypoxia-induced renal damage and functional deterioration. It is implicated in the clearance of calcifying nanoparticles, mitigation of inflammation, attenuation of fibrotic tissue remodeling, and polarization of macrophages. (I) In normoxia, liver-derived fetuin-A (green) locally binds calcium in mineral particles (light gray), inhibits their maturation and accumulation (dark gray). Macrophages exhibit an M2 anti-inflammatory phenotype (M2 MΦ, blue). (II) The relatively low abundance of fetuin-A-free calcium biominerals (dark red) in normoxic conditions is not sufficient to elicit a noticeable inflammatory response. The majority of macrophages is M2 polarized, only a small fraction shows a M1 pro-inflammatory phenotype (M1 MΦ, magenta). (III) In hypoxia, increased tissue damage leads to calcium mineral overload and a polarization shift towards M1 MΦ. However, local induction of fetuin-A (yellow) augments the clearance of calcium biominerals and keeps the polarization of M1 MΦ in check, thus preventing inflammation and fibrotic remodeling. (IV) Fetuin-A deficiency in an hypoxic environment leads to an unbalanced accumulation of calcium biominerals and their unchecked maturation (orange), which overwhelms the intrinsic clearing capacity of the tissue. This further promotes the polarization of pro-inflammatory M1 MΦ, and culminates in inflammation (red) and fibrotic tissue remodeling.

macrophages (magenta), inflammation, and TGF-β mediated fibrotic tissue remodeling, which, in the long-term, promotes proteinuria and decreased GFR in adult offspring. This model principally also applies to IRI in adult kidneys, with the difference that the renally produced fetuin-A is substituted by externally administered fetuin-A.

Beyond its established role as a calcification inhibitor in the serum phase, we propose a broader view of fetuin-A: as a potent calcium mineral scavenger, it counteracts ectopic pathological tissue calcification, especially during hypoxic stress, emerging thus as an important "gate keeper" of tissue integrity. In the developing organism, it plays an important role in fetal programming according to Barker's hypothesis; in adults, it is a potential therapeutic modality counteracting renal tissue remodeling upon ischemic reperfusion injury. These properties of fetuin-A have a high translational potential in human medicine, as it is not restricted to the treatment of hypoxia/ischemia-related injury, but could also be applicable as a replacement therapy in conditions associated with fetuin-A depletion such as CKD or acute inflammation.

## Methods

**Animals**. Breeding, genotyping and all animal experiments were conducted according to the Swiss law for the welfare of animals and were approved by the local authorities (Canton of Bern BE96/11, BE105/14, and BE105/17). All mice, including *Ahsg*^tm1Mbl^ mice and *Clcn5*^tm1Gug^ mice (kindly provided by Prof. Olivier Devuyst, University of Zurich) were maintained on a C57BL/6 background. All mice were housed in IVC cages with free access to chow and water and a 12 h day/night cycle, 23 °C ambient temperature and 40–60% humidity.

For timed matings, females in breeding were checked for vaginal plugs every morning, and if present the time point was set to gestational day (E) 0.5. *Ahsg* KO

mice were obtained from heterozygous breeding pairs, also giving rise to heterozygous and wt littermates that were used as controls. For induction of hypoxia, E13.5 pregnant mice were transferred into a hypoxic glove box (Coy Laboratory Products, Grass Lake, USA). The next day, the oxygen content was gradually lowered to 10% within 6–8 h with intermittent pauses at 16% and 12.5% to acclimatize the animals to the increasing hypoxic conditions. An electric fan inside the chamber maintained adequate air circulation. The $CO_2$ level was kept low by chelating excess $CO_2$ in soda lime (Sigma, 72073) filled cartridges connected to the air circulation system. Excess humidity was absorbed by silica gel orange granulate (Sigma, 1.01969), changed every day. Daily food consumption (weight of food initially provided minus the weight of food remaining after 24 h) and maternal weight were recorded from E0.5 to E18.5 for each mouse. From these data, the average daily food consumption was calculated as a fraction of body weight. Furthermore, to calculate the total amount of consumed food, the area under the curve (AUC) was determined. Until E14.5, all dams had free access to food. From E14.5 to E18.5, dams of the caloric control group received an constrained diet, which was the same fraction of food that was consumed by the animals in the hypoxic group. Pregnant hypoxic or control mice were euthanized on E18.5 and fetuses and placentas were collected, weighed and prepared for further analysis. Fetal kidneys were dissected in PBS using a Leica M80 stereoscope. Assessment of renal functions (GFR and proteinuria) were performed in 9 weeks old fetal hypoxic or normoxic mice.

For ischemia-reperfusion injury (IRI), surgery was performed in 10–12-week-old wt mice Briefly, IRI was induced in the left kidney by clamping the renal vessels for 30 min, the right kidney served as a control. Immediately after surgery, mice received the first treatment of either physiological NaCl solution or bovine Fetuin-A (100 µg/g body weight) via intraperitoneal injection. Injections were repeated daily for four days. Kidneys were isolated on day 5 and analyzed.

**Transcutaneous assessment of glomerular filtration**. The glomerular filtration rate (GFR) was determined in conscious animals as previously described[89]. Briefly, the plasma clearance of FITC-sinistrin (Fresenius-Kabi, LI9830076) is measured across the skin using light-emitting diodes with an emission maximum for FITC at 470 nm and a photodiode detecting the fluorescent light with a maximum sensitivity at 525 nm. The decrease in fluorescence intensity over time is then converted into GFR.

**Proteinuria**. Urine protein content was determined using the Bradford Assay. Three microliters of urine and 150 μl of 1x Bradford reagent were mixed, incubated at RT for 5 min and absorbance was measured at 595 nm. (5x Bradford reagent was prepared by dissolving 50 mg Brilliant Blue G-250 (Sigma, B-1131) in 24 ml ethanol and 50 ml 85% phosphoric acid, the adjusting the total volume to 100 ml with ultra-pure water). Urine creatinine content was determined using the Jaffe method. 10 μl of 1:10 diluted urine was mixed with 100 μl Creatinine reagent, incubated at RT for 10 min and absorbance was measured at 510 nm. (Creatinine reagent consisted of 10 mM picric acid and 250 mM NaOH, pH 13). Finally, the protein creatinine ratio was calculated for each sample.

**Glomerular count**. One-hundred micrometers Z-stack images of whole-mount E18.5 kidneys stained for nephrin (R&D AF3159) were analyzed with the open source image processing software Fiji (ImageJ, version 2.0.0-rc69/1.52i, https://imagej.net/Fiji). In the TrackMate v3.8.0 plugin, the Downsample LoG detector was set to 80.0 pixel for the estimated blob diameter with a 16-pixel threshold and downsampling factor 2. The number of spots per frame were added to calculate the number of glomeruli per kidney. A 100 μm distance between frames was chosen to avoid double counting of identical glomeruli in consecutive images, given an average glomerular diameter of 80 μm.

**Flow cytometry**. Fetal kidneys were dissociated with collagenase I (2 mg/ml in 1% BSA/PBS; C9891, Sigma) for 20 min at 37 °C. The cell suspension was filtered through a 40 μm pore size filter, washed and then stained using a panel of antibodies for 30 min at 4 °C. Cells were sorted on a SORP LSRII using FACSDiva and analyzed with FlowJo 10.6.1. Antibodies are provided in Supplementary Table 7.

**Microarray analysis**. Total RNA from male hypoxic, normoxic, and caloric control group E18.5 kidneys was isolated using the RNeasy Mini Kit (Qiagen, 74104). Only high-quality RNA (RIN > 8, 260/280 ratio > 2, 260/230 ratio > 1.8) was used for further analysis. One-hundred nanograms of total RNA samples were processed with the Ambion® WT Expression Kit (4411973, life technologies). In all, 5.5 μg of the complementary DNA (cDNA) was fragmented and labeled with GeneChip® WT Terminal Labeling kit (901525, Affymetrix). In all, 2.3 μg biotinylated fragments were hybridized to Affymetrix Mouse Gene 1.0 ST arrays at 45 °C for 16 h, washed and stained according to the protocol described in Affymetrix GeneChip® Expression Analysis Manual (Fluidics protocol FS450_0007). The arrays were scanned with Affymetrix GeneChip® Scanner 3000 7G and raw data was extracted from the scanned images and analyzed with the Affymetrix Power Tools software package. Hybridization quality was assessed using Affymetrix Expression Console software (version 1.1.2800.28061). Normalized expression signals were calculated from Affymetrix CEL files by the Robust Multi-array Average algorithm (RMA). Differential hybridized features were identified using the R Bioconductor package "limma" that implements linear models for microarray data[90]. P-values were adjusted for multiple testing with Benjamini and Hochberg's method[91] to control the false-discovery rate (FDR). Probe sets showing at least 1.3-fold-change and a FDR < 0.05 were considered significant. Differential expression values between hypoxic, normoxic, and caloric control group E18.5 kidneys were mapped with Heatmapper[92] (http://www.heatmapper.ca) using average linkage and Euclidean distance measurement. For functional annotation, GO-term analysis was performed using the DAVID platform[93] (https://david.ncifcrf.gov).

**RT-qPCR**. Total RNA was isolated using TRIzol® reagent (Invitrogen 15596026) according to the manufacturer's protocol. RNA concentration and quality was determined with a Nanodrop 1000 spectrophotometer (ThermoFisher Scientific, Switzerland) and 1000 ng were transcribed into cDNA using PrimeScript RT Reagent Kit (Takara, RR037A). cDNA was diluted to 2 ng/μl and qPCR was performed with either TaqMan Gene Expression Assays (ThermoFisher) or FAM-labeled UPL probe (Roche) plus corresponding gene-specific primers and TaqMan Fast Universal PCR Master mix (Applied Biosystems, 4352042) on a 7500 Fast Real-Time PCR System (Applied Biosystems). Data analysis was performed with Microsoft Excel. The $2^{(-\Delta\Delta Ct)}$ method was used to calculate the relative expression levels for RT-qPCR. Primer and probe sequences or Assay IDs are listed in Supplementary Table 4.

**Identification of HIF-binding sites**. For the identification of putative HIF-binding sites, the 20 kb genomic region overlapping with the *Ahsg* locus (10 kb up- and downstream of the start codon) of 15 different species (cat, chicken, chimp, cow, dog, ghost shark, horse, human, mouse, pig, rabbit, rat, sheep, xenopus, zebrafish) was analyzed with the JASPAR database[94] (http://jaspar.genereg.net). The relative profile score threshold was set to 90%. For enrichment analysis, putative sites of all species were clustered in 1 kb windows. The identified sites are listed in Supplementary Table 5. HREs with a relative score >0.9 are shown in red, >0.93 in orange and >0.97 in yellow.

For the alignment of human HIF alpha sites identified by ChIP-seq of hypoxic MCF7 cells[29] (https://www.ncbi.nlm.nih.gov/geo/query/acc.cgi?acc=GSE28352) with active regulatory marks of the *AHSG* locus, the HIF-1-alpha and HIF-2-alpha data sets encompassing chr3: 180,000,000–190,000,000 (including the *AHSG* locus) were converted into BAM files using the web-based Galaxy platform[95]

(https://usegalaxy.org) and uploaded to the human assembly GRCh37/hg19 on the USCS genome browser[96] (https://genome.ucsc.edu). To this alignment, the following data sets were added: layered chromatin marks often found near active regulatory elements of seven cell lines (H3K27Ac and H3K4Me1, ENCODE) and open chromatin of hypoxic MCF7 cells (DNaseI HS, ENCODE). For the mouse *Ahsg* locus, the potential HIF-binding sites identified with JASPAR were aligned with data sets (DNaseI HS, ENCODE/UW; H3K27Ac and H3K4Me1, ENCODE/LICR) derived from 8 weeks mouse liver, heart and kidney, showing the mean signal intensity (bar graphs, auto-scaled, log-transformed, smoothened (16 pixels)).

**Molecular cloning**. For luciferase assays, the 2.5 kb promoter fragment upstream of the mouse *Ahsg* ATG was amplified from genomic DNA (C57BL/6) using specific primers and PrimeSTAR® GXL DNA Polymerase (Takara, R050A). The 500 bp promoter fragments (wt and mutant) and the 500 bp fragment of intronic sequences (wt and mutant) were synthesized by IDTDNA (https://eu.idtdna.com/pages). The promoter fragments were inserted into the pGL3-basic-P2P-607 plasmid (kind gift of Prof. Wenger, Zurich) with *Nco*I and *Sac*I restriction enzymes (both NEB, R3193S, and R3156S). Intronic fragments were inserted into the pGL3-basic vector or the pGL3-basic vector carrying the promoter fragments using *Bam*HI and *Sal*I restriction enzymes (both NEB, R3136S and R3138S). Primer and fragment sequences are listed in Supplementary Table 6.

For in situ hybridization, the cDNA of exons 2–5 of mouse *Ahsg* was obtained from IDTDNA and cloned into pBluescriptII KS- using SpeI and *Eco*RI restriction enzymes (both NEB, R3133S and R3101S).

**Luciferase assay**. Twenty-four hours after transfection, cells were washed twice with PBS, lysed (250 mM KCl, 50 mM Tris/H₃PO₄ pH7.8, 10% glycerol, 0.1% NP40) on ice for 20 min and centrifuged at full speed $(17,000 \times g)$ at 4 °C for 10 min. Of the supernatant, 10 μl were used for each reaction. Injection of reaction solutions (Luciferase: 100 μl of 25 mM Tris/H₃PO₄ pH7.8, 10 mM MgSO₄, 2 mM ATP pH7.5, 50 μM luciferin; Renilla: 100 μl of 50 mM Tris/HCl pH7.6, 100 mM NaCl, 1 mM EDTA, 0.5 μM coelenterazine) and activity measurement was performed with a Fluoroskan Ascent FL (ThermoFisher). Each sample was measured in duplicates and luciferase activity was normalized by renilla activity.

**Whole-mount in situ hybridization**. E18.5 kidneys were fixed in 4% PFA, dehydrated and stored in methanol at −20 °C. In situ hybridization using a digoxigenin-labeled riboprobe was performed as described[97]. Probes were generated using the DIG RNA Labeling Mix (Roche, 11175025910) and T3 or T7 RNA polymerase (both Roche, 11031163001 or 10881767001). An alkaline phosphatase-conjugated antibody was used to detect the DIG-labeled probes (Roche, 11093274910).

**TUNEL staining**. Fragmented DNA in apoptotic cells was detected using the Promega DeadEnd Colorimetric TUNEL System (G7360) according to the manufacturer.

**Histochemistry**. For immunohistochemistry, PFA-fixed, paraffin-embedded tissue sections were rehydrated and endogenous peroxidase was blocked by incubating the slides in 1.5% H₂O₂ solution (0.02 M citric acid, 0.06 M Na₂HPO₄) at RT for 15 min in the dark. Antigen retrieval was performed by boiling in Tris-EDTA buffer pH 9 for 20 min followed by slow cool down to RT. After blocking in 2% BSA in PBS at RT for 1 h, the sections were incubated with primary antibodies in blocking solution o/n at 4 °C. Antibodies were diluted according to the recommendation of the manufacturer. Following three washing steps in PBS, the sections were incubated with HRP-conjugated secondary antibodies (mouse or rabbit: Dako EnVision+ System from Agilent (K4001 or K4003), ready-made solution, no dilution required; goat: SCBT, sc-2304 1:5000) for 1 h at RT. After three washing steps in PBS, the signal was developed with DAB (Agilent, K3468). The sections were counterstained with Harris haematoxylin solution (Sigma, HHS16), dehydrated and mounted using Eukitt medium (Sigma, 03989).

For Picrosirius red staining of collagen, de-waxed, rehydrated tissue sections were incubated in staining solution (0.5 g Direct Red 80 in saturated aqueous solution of picric acid (both Sigma, 365548 and P6744)) for 1 h at RT. After washing twice in acidified water (0.5% glacial acetic acid), the sections were dehydrated and mounted using Eukitt.

**Immunofluorescent staining**. Cryosections were fixed in 4% PFA at RT for 10 min, washed twice in PBS and permeabilised by incubation in PBST (0.1% Triton X-100 in PBS) at RT for 10 min. After blocking in 10% FCS, 0.5% Tween-20 in PBS at RT for 1 h, the sections were incubated with primary antibodies in blocking solution o/n at 4 °C. Following three washing steps in PBS, the sections were incubated with fluorescence-conjugated secondary antibodies in blocking solution in the dark for 1 h at RT. Antibodies were diluted according to the recommendation of the manufacturer. DNA was stained with DAPI 1:5000 in PBS. Sections were mounted in MOWIOL solution (2.4 g MOWIOL 4–88 reagent (Merck, 475904) in 6 g glycerol and 18 ml 0.13 M Tris pH 8.5). Primary and secondary antibodies are provided in Supplementary Table 7.

For whole-mount immunofluorescence staining of E18.5 kidneys, the iDisco staining protocol[98] (https://idisco.info/idisco-protocol/) with methanol pre-treatment was applied. An incubation time $n = 1$ day and a solution volume of 1.6 ml was used for the relevant steps. The kidneys were mounted in 8-well glass chamber slides (ThermoFisher, 154534) and imaged immediately.

**Fluorescent detection of calcium**. Thick cryosections (30 μm) were incubated with 10 ng/ml ATTO 488 fluorescently labeled fetuin-A (in calcium-free PBS) in the dark at RT for 60 min, rinsed three times with PBS and mounted with MOWIOL solution. Nuclei were counterstained with DAPI.

**Imaging**. Fluorescence imaging was performed on a IMIC digital microscope (FEI, Type 4001) using the Polychrome V light source, an Orca-R² camera controller from Hamamatsu (C10600) and Live Acquisition software (FEI, version 2.6.0.14). Image analysis was performed using Offline Analysis software (FEI). Bright-field imaging was performed on a Nikon E600 microscope equipped with Nikon objectives (Plan Fluor ELWD 20x/0.45, Plan Apo 40x/1.0 Oil and 60x/1.40 Oil) using a Digital Sight DS-UE camera controller and DS-Ri1 camera (both Nikon). Image analysis was performed using Nikon software NIS Elements 4.0.

**Enzyme-linked immunosorbent assay (ELISA)**. Plasma fetuin-A levels of E18.5 fetuses were determined using the Mouse Fetuin-A/AHSG Quantikine ELISA Kit (R&D, MFTA00) according to the manufacturer.

**Cell culture**. The normal rat kidney (NRK) cell line was cultured in Dulbecco's modified Eagle medium ((DMEM) (Gibco, 41965-039)) and 10% fetal bovine serum (FBS). The human kidney (HK-2) cell line was cultured in Keratinocyte-SFM medium (Gibco, 17005-075). For luciferase assays, NRK cells were transfected with luciferase reporter plasmids and pCMV-Renilla (10% of total transfected DNA, used for normalization) using jetPrime® reagent (Polyplus, 114-07), stimulated with 1 mM DMOG (Echelon Biosciences, F-0010) 6 h after and harvested 24 h after transfection. For hypoxia, cell culture was performed at 0.2% oxygen for 48 h.

Primary proximal tubular cells (pPTC) were isolated from 3–4-week-old kidneys as previously described[99]. Briefly, proximal tubular fragments were obtained by digesting cortical kidney tissue with collagenase and subsequent filtration through a 250 μm followed by an 80 μm pore size membrane. pPTCs were cultured in DMEM/F12 (Gibco, 21041-025) supplemented with 15 mM HEPES, 0.55 mM NaPyruvate, 1% NEAA and renal epithelial cell growth medium (REGM) supplements (Lonza, CC-4127). Instead of FBS, serum from *Ahsg* KO mice was used. Upon confluency, pPTCs were split using Accutase solution (Sigma, A6964) and replated once. For hypoxia, cell culture was performed at 0.2% oxygen for 48 h. For treatment, 100 μg/ml Fetuin-A (Sigma, F3385) or bovine serum albumin (BSA, Sigma, A3059) was added to the culture medium 48 h before the end of the experiment. Before treatment with 5 ng/ml rmTGF-β1 (R&D, 7666-MB) for 5 min, cells were starved for 24 h.

**Western blot analysis**. Total protein lysates were obtained using RIPA buffer (Sigma, R0278) supplemented with protease inhibitors (Roche, 11836153001). Proteins were separated by sodium dodecyl sulfate polyacrylamide gel electrophoresis and blotted onto polyvinylidene fluoride membranes (ThermoFisher, 88518). Upon blocking with 5% milk in TBST, the membranes were incubated with primary antibodies at 4 °C o/n. Antibodies were diluted according to the recommendation of the manufacturer. Incubation with HRP-conjugated secondary antibodies (dilution 1:5000) was performed at RT for 1 h. The signal was detected with ECL (GE Healthcare, RPN2106) or SuperSignal (ThermoFisher, 34076) depending on signal intensity. Densiometric analysis was performed with the open source image processing software Fiji (ImageJ, version 2.0.0-rc69/1.52i, https://imagej.net/Fiji). Antibodies are listed in Supplementary Table 7.

**Data analysis**. Statistical analysis and graphs were performed with Prism 8 (https://www.graphpad.com). If not specified otherwise, two groups were compared by unpaired 2-sided *t*-tests, multiple groups by ordinary one-way ANOVA with Tukey's multiple comparisons test. ****$P < 0.0001$; ***$P < 0.001$; **$P < 0.01$; *$P < 0.05$; ns, not significant; Data are presented as mean ± SEM, unless specified otherwise.

**Reporting summary**. Further information on research design is available in the Nature Research Reporting Summary linked to this article.

## Data availability
Microarray raw data were deposited at the GEO genomics data repository under the accession code GSE148778. HIF ChIP-seq data sets are accessible under GSE28352. Other genome-wide dataset were obtained from UCSC Genome Browser at https://genome.ucsc.edu (for human: H3K27Ac, H3K4Me1, DNaseI HS (ENCODE); for mouse DNaseI HS (ENCODE/UW),H3K27Ac and H3K4Me1 (ENCODE/LICR)) or JASPAR database at http://jaspar.genereg.net (for potential mouse HIF-binding sites). All other

data supporting the findings of this study are available from the corresponding author on request. Source data are provided with this paper.

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

## Acknowledgements

We thank D. Lambert for performing GFR measurements; P. Näf and A. Karolin for their help with flow cytometry; K. Harshman (Geneva, Switzerland) for performing micro-arrays; J. Loffing (Zurich, Switzerland) for giving antibodies; O. Devuyst (Zurich, Switzerland) for generously providing *Clcn5* KO mice and fruitful discussions; R. Wenger for giving luciferase reporter plasmids and thoughtful comments on the manuscript. This work was supported by the NCCR (National Center of Competence in Research) Kidney. CH grant of the Swiss National Science Foundation to U.H.-D., EU Grants (FP7-PEO-PLE-COFUND-2008) for the Postdoctoral Program in Integrative Kidney Physiology & Pathophysiology (IKPP) to S. Rudloff and M.J. (246539) and IKPP2 to K.D. (608847).

## Author contributions

S. Rudloff, M.J., S. Rodriguez, and K.D. performed experiments. S. Rudloff, M.J., S. Rodriguez, K.D. and U.H.-D. planned and analyzed experiments. S. Rudloff, W.J.-D. and U.H.-D. wrote and edited the manuscript. U.H.-D. conceived the study.

## Competing interests

The University of Bern has filed a European patent on behalf of U.H.-D. and S. Rudloff with application number: EP20162490. The patent relates to use of fetuin-A in treating renal disorders. The other authors declare no competing interests.

## Additional information

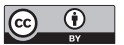

