## [Peer Review File · Nature Communications]

Reviewers' comments:

Reviewer #1 (Remarks to the Author):

Comments on the Ms entitled "Fetuin-A, a novel HIF target, safeguards tissue integrity during hypoxic stress" (NCOMMS-19-15730)

Ms submitted by Stefan Rudloff et al. (Corresponding author U. Huynh-Do)

In this Ms the authors have investigated the role of FetuinA (ahsg) on hypoxia induced kidney impairment; it protects kidney tubule tissue damage through scavenging calcium mineral. According to the authors FetA which is a target gene of HIF, helps to prevent ectopic calcification to maintain integrity of tissues, thus protecting intra-uterine growth restriction (IUGR) kidneys from progression to CKD. FetuinA also helps in protecting hypoxia induced kidney damage by inhibiting macrophage infiltration and fibrotic remodelling. However, following points are necessary to be considered by the authors.

1. The authors have demonstrated the model of chronic foetal hypoxia by exposing pregnant mice to 10% oxygen followed by all experiments with foetus obtained from these hypoxic and control mice. The level of FetA in foetal proximal kidney tubule was their major concern as hypoxic stress induced FetA prevents progression to Chronic Kidney Disease (CKD). According to them increase of FetA due to hypoxia was not reflected in the blood but they did not measure level of FetA in the mother and also the preventive role of FetA in the mother kidney tubule. Since hypoxia stress was exposed to mother, a relationship between the mother and foetus is very important. This aspect is important because hypoxic stress on mother was transmitted to the foetus.
2. Authors have shown that in ahsg KO mice there is an increased population of F4 /80+ cells suggesting macrophage accumulation. But it is reported that FetuinA (FetA) acts as a chemoattractant for macrophage. Have authors examined this aspect? In addition it is necessary also to find how much of F4 /80+ population is converted to CD11c+ as these are important in evaluating the inflammatory status which usually occurs in presence of FetA.
3. The authors have stated that in hypoxic condition FetA is expressed only in the proximal tubule (PT), but from the fig 3(i-l) it appears that FetA is also considerably expressed in all other parts of the kidney tubule such as TAL,DCT,CD. Hence, it is important to examine whether hypoxia induced HIF and FetA are also involved in these portions of the kidney tubule in a similar manner as found with PT.
4. It has been mentioned in fig 2f that the authors could not detect a significant rise in foetal plasma FetuinA levels among hypoxic, normoxic and caloric control group but the ELISA data reveals mean value difference of about 250 microgram/ml and this difference has been found to be statistically insignificant. However, such a rise in the mean value of FetA in serum due to hypoxic stress seems to be meaningful.

Reviewer #2 (Remarks to the Author):

The authors detected Fetuin A as a protective factor against CKD, a factor that is induced by HRE, a downstream target of HIF-1alpha. They studied it in gestational hypoxia-induced mouse offspring's kidney early in life and subsequently as an adult. In addition, they employed in-vitro conditions with various cell lines to examine the biological role of Fetuin A and its interaction with the TGF-beta related phosphorylation of SMAD3. Finally, they employed mice lacking Fetuin A to examine the impact in-vivo and in ex-vivo conditions. Fetuin A is well described in the literature in various conditions, known to exert its anti-inflammatory effects and causing calcium-related damage to joints, heart valves, brain vasculature and so on. These authors have attempted to examine its role in hypoxia-induced IUGR.

The issues of concern are the following:

1. While the authors acknowledge the limitations of their model related to reduced food intake by the dams, it is not clear how much of a reduction this truly is? That is not mentioned in the manuscript. It is important to state this, given that most studies have shown that food intake reduction by the dams leads to IUGR in the offspring evident at birth of the pups. The fact that they did not see this, is concerning to this reviewer. It is established that IUGR due to gestational hypoxia is related to a reduction in maternal food intake. This needs more clarification than simply stating that their food intake reduced group did not show IUGR.
2. While the number of pups/litter at birth were 6 in all three groups, it is important to ensure this number stays constant throughout the suckling phase if the adult studies are to mean something. This is not shown. Usually, the the C56/BL6 background dams cannibalize their IUGR pups, and the fact that this is not mentioned is a weakness of the study. If postnatal nutrition is variable between the different litters, the comparison will not stand on an equal footing.
3. It would be important to take on EM studies with immunogold to demonstrate the changes with Fetuin A, and its localization. This is also important in the case of the KO mice employed in this study.
4. Figure 4, blots are simply shown. It is important to provide quantification of these blots as well.
5. No power analysis is provided towards pre-determining the sample size for these studies. It appears that the sample size was arbitrarily chosen. This is a major weakness of this paper.
6. It was not clear if males or females were examined, since it is critically important to study both sexes, to determine if this phenomenon is sex-specific, or affects both.
7. While the discussion mentions other organs, it is important to have included another organ system in which such changes are seen or not seen, to determine if these observations are kidney specific, and not a generalized phenomenon in the hypoxia-induced IUGR offspring.

Reviewer #3 (Remarks to the Author):

In their study Rudloff and colleagues using a valid murine model of fetal hypoxia-induced intra-uterine growth restriction (IUGR) at first instance characterized its short and long term consequences on the kidney. Importantly applying this hypoxic model in the Ahsg KO, i.e. fetuin-A KO mice revealed the highly interesting finding that fetuin-A, being a well-known calcification inhibitor, also acts as an evolutionary conserved HIF (hypoxia-inducible transcription factor) target gene, by which it protects IUGR kidneys from fast progression to CKD through clearance of calcium particles thereby attenuation microphage infiltration, inflammation and fibrosis. This is an interesting study shedding new light on fetuin-A function which on the longer run might also have therapeutic potential not only in the field of CKD but also in other hypoxia-mediated disease states.

The study set-up of the present study is straightforward and very well outlined with various in vivo as well as in vitro studies following up each other in a logic way, thereby confirming and proving evidence to each other.

The authors propose a nice mechanistic model (Figure 8). Although the proposed model sounds convincing, going through the paper, this reviewer leaves a bit with a 'the chicken and the egg' story in that one may ask the question if fetuin (i) directly protects the kidney against hypoxia-induced renal damage thereby inhibiting ectopic calcification or (ii) through inhibition of calcium mineral deposition/maturation rescues the kidney from damage and ensuing CKD. I would not see this comment as a negative point, however, would propose to deal herewith in the discussion section of the paper. In line herewith, one should put the potential effect (potential therapeutic use) of fetuin A on hypoxia-induced renal damage (perhaps other organs) in the setting of e.g. ischemia/reperfusion acute kidney injury in which, to the best of this reviewer's knowledge no increased calcium deposition has been demonstrated so far, in perspective.

Some minor comments:

Figures, Graphs: Data in figures alternately are presented as either individual values, box-whisker plots, average with SD bars. Perhaps it might be wise to present data in a way that to a certain extent covers the various ways of presentation; i.e. individual data presented as dot plots next to

the average for the group with appropriate error bars.

The number of animals included for the various experiments/analyses is not always well defined; e.g. ≥ 5 or ≥ 4 . I would propose that the exact number should be mentioned.

Introduction:

Last para: In case additional data on the characterization of the hypoxic model as well as the KO mice models are available in literature or unpublished, reference should be made to these in e.g. the reference list or supplemental material.

Results:

Maternal weight gain was significantly reduced in hypoxic dams going along with a decreased LBW and number of nephrons in the hypoxic fetuses, it might be worth the effort to calculate the LBW/maternal weight and number of nephrons/LBW ratio's in the various groups and include these data in the supplementary information.

There was a reduction in the number of nephrons. Did the authors also have the possibility to check the intactness of the nephrons; e.g. was there a difference in the incidence of e.g. atubular glomeruli.

Page 7: lines 131 - 135: '... fetuin was expressed only in the PT of hypoxic fetal kidneys ...'. This cannot be derived from figure 3i-I as data from normotoxic kidneys are not presented.

Page 9, line 191: 189 – 192: '... positive 488-FA staining in the absence of von Kossa or Alizarin-Red ... structures merely enriched with calcium ... calcium-phosphorus aggregates ...' Has this statement been evidenced in literature. If so reference should be made to it.

Page 9, line 195: this sentence sounds rather speculative.

Page 9, line 195, figures 6b,c: No staining is shown for the normotoxic Ahsg KO's. The same counts for Figure 7. Was here any reason for this.

Page 10, line 214: As effects on Tgfb1 are an essential part in the reasoning I would propose to include the Tgfb1 figure in the figures of manuscript rather than including it in the supplementary information

Page 10, lines 215 -218: I would propose to turn around this sentences; i.e. '... fibrotic markers were generally enhanced in hypoxic culture conditions and further increased in Ahsg KO pPCTCs ...'

Page 210, line 224: Could we, as an alternative name this a 'soluble decoy receptor protein' also.

Discussion:

Page 12, 251-251: In order to put the model in a clinical perspective it might be of interest to translate the degree of renal insufficiency in the mouse model to a certain CKD .. stage in humans.

Methods:

Page 16: Was there any mortality in the animals?

Page 19, RT-qPCR: Which housekeeping gene was used?

Patrick C. D'Haese

Point-to-point rebuttal to the Reviewers' comments

First, we would like to express our deepest thanks to all reviewers for their thoughtful, precise and very helpful comments. We have addressed them in the revised version of our manuscript and believe that they greatly improved the quality of our study. In the following passages, our response to each reviewer is giving point by point.

Reviewer #1 (Remarks to the Author):

Comments on the Ms entitled "Fetuin-A, a novel HIF target, safeguards tissue integrity during hypoxic stress" (NCOMMS-19-15730) Ms submitted by Stefan Rudloff et al. (Corresponding author U. Huynh-Do) In this Ms the authors have investigated the role of FetuinA (ahsg) on hypoxia induced kidney impairment; it protects kidney tubule tissue damage through scavenging calcium mineral. According to the authors FetA which is a target gene of HIF, helps to prevent ectopic calcification to maintain integrity of tissues, thus protecting intra-uterine growth restriction (IUGR) kidneys from progression to CKD. FetuinA also helps in protecting hypoxia induced kidney damage by inhibiting macrophage infiltration and fibrotic remodelling.

However, following points are necessary to be considered by the authors.

1. The authors have demonstrated the model of chronic foetal hypoxia by exposing pregnant mice to 10% oxygen followed by all experiments with foetus obtained from these hypoxic and control mice. The level of FetA in foetal proximal kidney tubule was their major concern as hypoxic stress induced FetA prevents progression to Chronic Kidney Disease (CKD). According to them increase of FetA due to hypoxia was not reflected in the blood but they did not measure level of FetA in the mother and also the preventive role of FetA in the mother kidney tubule. Since hypoxia stress was exposed to mother, a relationship between the mother and foetus is very important. This aspect is important because hypoxic stress on mother was transmitted to the foetus.

The hypoxic environment is very likely to have an influence on the mother as well, however in our study we were primarily focussed on investigating the consequence of hypoxia-induced IUGR on the developing kidney of the fetus. On the other hand, we also believe that shedding more light onto the maternal phenotype is important to support our findings. Thus, we have determined the level of fetuin-A in mother animals (Fig. R1-1A, also new Supplementary Fig. 4a) and also looked at the histology of normoxic and hypoxic maternal kidneys using HE staining (Fig. R1-1B). As mentioned in the paper, the hypoxic mothers behaved pretty normally. We found no significant differences in the serum level of fetuin-A between normoxic and hypoxic mothers at the end of pregnancy (E18.5), nor could we detect changes in tubular morphology. Although a further corroboration of the maternal phenotype would be interesting to pursue, we think that it would require a whole new set of experiments that are definitely out of scope in relation to the presented data on the fetal outcomes of chronic hypoxia and their long-term consequences.

Fig. R1-1. Effect of chronic hypoxia on mothers. (A) Serum fetuin-A levels are not changed between hypoxic or normoxic mothers. (B) Histological examination did not reveal changes in renal morphology between normoxic or hypoxic mothers.

2. Authors have shown that in *Ahsg* KO mice there is an increased population of F4/80+ cells suggesting macrophage accumulation. But it is reported that FetuinA (FetA) acts as a chemoattractant for macrophage. Have authors examined this aspect? In addition it is necessary also to find how much of F4/80+ population is converted to CD11c+ as these are important in evaluating the inflammatory status which usually occurs in presence of FetA.

Indeed, we observed an accumulation and clustering of F4/80+ cells in *Ahsg* KO kidneys derived from hypoxic fetuses. Indeed we are aware of the paper from Chatterjee et al., reporting fetuin-A to be a chemoattractant for macrophages, for instance in adipose tissue [1]. However, in *Ahsg* KO mice there is no fetuin-A (Figs. 6k'' and m''), since it is knocked out and thus, it cannot act as a chemoattractant in this setting. On the other hand, hypoxia, a well-known inducer of inflammation, promotes cytokine release and therefore is involved in the homing of cells of the innate immune system including macrophages [2]. Moreover, the loss of fetuin-A even aggravated the inflammatory response and in its wake an enhanced infiltration of F4/80+ cells, confirming its role as an anti-inflammatory protein due to its role in the removal of mineral debris in multiple pathological settings [3].

3. The authors have stated that in hypoxic condition FetA is expressed only in the proximal tubule (PT), but from the fig 3(i-l) it appears that FetA is also considerably expressed in all other parts of the kidney tubule such as TAL, DCT, CD. Hence, it is important to examine whether hypoxia induced HIF and FetA are also involved in these portions of the kidney tubule in a similar manner as found with PT.

In Figure 3, we provided evidence that fetuin-A (in green) was only expressed in the PT and not in other parts of the kidney tubule including the TAL, DCT and CD. To show this, we used different marker proteins (in red) of these 4 renal tubule segments (Aqp1 – aquaporin 1 for the PT, Nkcc2 for the TAL, Ncc for the DCT and Aqp2 – aquaporin 2 for the CD). The only marker protein that colocalized with fetuin-A is Aqp1 (Fig. 3i''), demarcating the PT. All other markers did not colocalize with Fetuin-A, instead they were found adjacent to Fetuin-A. This is due to the anatomy of the nephron with convoluted and straight segments and its looping into the medulla and back to the cortex. Figure R1-3 (also new Supplementary Fig. 4d) shows a fetal kidney stained for Fetuin-A (here in red), highlighting the cortical localization of Fetuin-A. No signal was detected in the medulla. We have included this image into our supporting information, because we realized that a good overview of the expression of fetuin-A showing an entire fetal kidney had been missing so far.

Fig. R1-3. Location of Fetuin-A in hypoxic fetal kidneys. Fetuin-A staining was present in the outer renal cortex, just below the nephrogenic zone (Supplementary Fig 4c), forming an arch running along the circumference of the kidney, exactly where mature PTs are located in an E18.5 kidney. No staining was found in the inner cortex or the medulla.

4. It has been mentioned in fig 2f that the authors could not detect a significant rise in foetal plasma FetuinA levels among hypoxic, normoxic and caloric control group but the ELISA data reveals mean value difference of about 250 microgram/ml and this difference has been found to be statistically insignificant. However, such a rise in the mean value of FetA in serum due to hypoxic stress seems to be meaningful.

We also regard the analysis of systemic Fetuin-A levels to be very relevant to our findings. Thus, we have repeated the ELISA experiments (Fig. R1-4, also new Fig. 2f), including more samples in each group, along with

the measurement of fetuin-A levels in the mother animals (Fig. R1-1A, also new Supplementary Fig. 3a). This new analysis confirmed our previous findings that no difference exists in systemic fetuin-A levels among normoxic, hypoxic, or Cc fetuses. We cannot fully explain the discrepancy in the absolute serum Fetuin-A values (now much lower than before), however a new ELISA kit with a different lot number was used.

Fig. R1-4. Serum levels of Fetuin-A in E18.5 fetuses. Serum Fetuin-A levels were comparable among normoxic, hypoxic, or caloric control E18.5 fetuses.

Reviewer #2 (Remarks to the Author):

The authors detected Fetuin A as a protective factor against CKD, a factor that is induced by HRE, a downstream target of HIF-1 α . They studied it in gestational hypoxia-induced mouse offspring's kidney early in life and subsequently as an adult. In addition, they employed in-vitro conditions with various cell lines to examine the biological role of Fetuin A and its interaction with the TGF-beta related phosphorylation of SMAD3. Finally, they employed mice lacking Fetuin A to examine the impact in-vivo and in ex-vivo conditions. Fetuin A is well described in the literature in various conditions, known to exert its anti-inflammatory effects and causing calcium-related damage to joints, heart valves, brain vasculature and so on. These authors have attempted to examine its role in hypoxia-induced IUGR.

The issues of concern are the following:

1. While the authors acknowledge the limitations of their model related to reduced food intake by the dams, it is not clear how much of a reduction this truly is? That is not mentioned in the manuscript. It is important to state this, given that most studies have shown that food intake reduction by the dams leads to IUGR in the offspring evident at birth of the pups. The fact that they did not see this, is concerning to this reviewer. It is established that IUGR due to gestational hypoxia is related to a reduction in maternal food intake. This needs more clarification than simply stating that their food intake reduced group did not show IUGR.

We completely agree with Reviewer #2 that food intake is highly relevant to judge the outcome of our study and regret that we did not explain the conditions for the caloric control (Cc) group in more detail. We have now presented our data with higher granularity as to the precise timepoints of food intake measurements, and in addition, performed numerous additional experiments (also to answer points 2 and 6) to address the raised concerns.

Food consumption (Fig. R2-1A, also new Fig. 1b): Hypoxic dams consumed 8.6 % less food than normoxic dams throughout the course of pregnancy, and this is why we introduced a second, "caloric control" group, where the amount of food was just reduced by 8.6 % in order to match the amount consumed by hypoxic dams. The reduced food intake was most pronounced within the first 48 hours of hypoxia (75 % reduction on day 1 and 50 % reduction on day 2 of hypoxia), the time period the mice seemed to need to adapt to hypoxia. However, on hypoxia day 3 and 4 food intake was back at 84 % ($p=0.04$) and 93 % (no significant difference), respectively. In comparison to other published protocols of food restriction [4, 5], where dams were not fed at all for up to 96 h before term [6], received only 50 % of ad libitum throughout gestation [7], or during the last 6 to 14 days before birth [8, 9], our 37 % food restriction from E14.5-18.5 (of which only 3 days are marked by nutritional restriction) is a very mild protocol.

Maternal weight gain (Fig. R2-1B, also new Fig. 1c): During pregnancy, normoxic dams gained 56.1 % weight, Cc dams 48.8 %, and hypoxic dams 33.1 %. The latter lost weight (9.1 %) during the first day of hypoxia, while the weight of caloric restricted dams remained constant in this time period (albeit they received the same amount of food), and that of normoxic dams increased by 7.3 %.

Fetal weight: Importantly, only fetuses from hypoxic dams, not however fetuses from the Cc group exhibited IUGR, fulfilling the criteria for small for gestational age (SGA) set by Lubchenco in 1963 [10], which is a birth weight 2 standard deviations (<3%) below the mean for gestational age.

In summary, these data reveal that our protocol of mild caloric restriction alone does not lead to IUGR, despite reduced food intake. Below, you can find the amended description of the diet in the methods and the main text, and the new images for Supplementary Figure 1 (Fig. R2-1).

Amended main text: [...] To model chronic fetal hypoxia, timed-mated pregnant mice were exposed to 10 % oxygen from E14.5 to E18.5 (Fig. 1a). **We observed that throughout the whole gestation dams in hypoxia ate 8.6% less compared to dams under ambient conditions (Fig. 1b), as a consequence of hypoxia exposure. For a better visualization, we plotted the differential food intake between these two groups as percent body weight and calculated the total consumed food as area under the curve (AUC). The reduced food intake was most pronounced within the first 48 hours of hypoxia, the time period the mice seemed to need to adapt to hypoxia. However, on hypoxia day 3 and 4 food intake was back at 84 % ($p=0.04$) and 93 % (no significant**

difference), respectively. Since caloric restriction itself is a known inducer of IUGR, we included a second control group in our analysis to exclude the possibility that our findings might not be due to hypoxia, but rather to reduced ingested calories. In this caloric control (Cc) group, normoxic dams were fed with an amount of food matching the amount of food consumed by the hypoxic mice (Fig. 1b). Thus, throughout gravidity hypoxic and Cc dams consumed 91.4 % of the food ate by normoxic dams, which is a very mild food restriction compared to the majority of published protocols¹²⁻¹⁴. During gestation, normoxic dams gained 56 % of body weight, while the weight increase for Cc or hypoxic was 49 % or 33 %, respectively (Fig. 1c), emphasizing the negative effect of hypoxia on weight gain. In more detail, the 7 % difference between normoxic and Cc dams was due to a zero net weight gain of Cc mice during the 24 hours of food restriction, thereafter weight gain normalized or was even higher than for normoxic dams. On the contrary, hypoxic dams lost 9.1 % of body weight during the initial 24 hours of hypoxia and it took 72 hours before their weight gain was back to control levels. Despite these differences, placental mass and the number of E18.5 fetuses per litter were indistinguishable among the three groups (Supplementary Figs. 1a,b). However and importantly, only hypoxic E18.5 fetuses showed LBW, fulfilling small for gestational age criteria¹⁵, whereas normoxic and Cc fetuses did not (Fig. 1d). [...]

Amended text in methods: [...] Excess humidity was absorbed by silica gel orange granulate (Sigma, 1.01969), changed every day. Daily food consumption (weight of food initially provided minus the weight of food remaining after 24 h) and maternal weight were recorded from E0.5 to E18.5 for each mouse. From these data, the average daily food consumption was calculated as a fraction of body weight. Until E14.5, all dams had free access to food. From E14.5 to E18.5, dams of the caloric control group received a constrained diet, which was the same fraction of food that was consumed by the animals in the hypoxic group. Pregnant hypoxic or control mice were euthanized on E18.5 and fetuses and placentas were collected [...]

Fig. R2-1. Time course and extent of reduced food intake and its effect on fetal mass. (A) Characterization of reduced food consumption in hypoxia. Daily food intake from day 1 to 19 of pregnancy shown in relation to the maternal body weight. Until the start of hypoxia (arrow, E14.5 or on day 15) the data comprises all dams, thereafter food intake is shown separately for normoxic dams (black dotted line) or hypoxic dams (bold dark grey line). During the course of pregnancy, normoxic dams consumed 292.2 % of their body, while hypoxic or caloric restricted mice ate 267 %. This corresponds to 91.4 % of the food consumed by normoxic mice. Time points of significantly different food intake are days 16-18. Multiple t tests; ****, $P < 0.0001$; *, $P < 0.05$. (B) Characterization of maternal weight gain. Daily weight gain from day 1 to 19 of pregnancy shown in relation to the maternal body weight. Until the start of hypoxia or caloric restriction (arrow, E14.5 or on day 15) the data comprises all dams, thereafter weight gain is shown separately for normoxic dams (black dotted line), hypoxic dams (bold dark grey line), or Cc dams (dashed light grey line). Time points of significantly different food intake are days 16-18. Multiple t tests; large asterisk ($P < 0.05$), normoxic vs. hypoxic dams; small asterisk ($P < 0.05$), normoxic vs. Cc dams; # ($P < 0.05$), hypoxic vs. Cc dams.

2. While the number of pups/litter at birth were 6 in all three groups, it is important to ensure this number stays constant throughout the suckling phase if the adult studies are to mean something. This is not shown. Usually, the the C56/BL6 background dams cannibalize their IUGR pups, and the fact that this is not mentioned is a weakness of the study. If postnatal nutrition is variable between the different litters, the comparison will not stand on an equal footing.

To address this important point, we repeated our experiments and recorded daily the number of pups per litter (males and females) and their weight for 28 days and thereafter weekly for another 4 weeks. Throughout the suckling phase, we did not detect significant changes in the size of the litters between normoxic or hypoxic mothers (Fig. R2-2A, also new Supplementary Fig. 1d), nor did the survival of the offspring differ significantly between these groups (Fig. R2-2B, also new Fig. 1f).

As to the issue of cannibalism, we did not find it to be a frequent finding in our study, but rather an exception. Often, the rate of cannibalism can vary between different mouse facilities or even among different rooms within the same facility, but this is obviously not a problem in our case, as our colleagues from different labs using the same mouse facilities did not report such a problem. If cannibalism occurred, it usually affected the weakest pup in the afflicted litters, regardless of the experimental condition. Thus, the surviving hypoxic offspring are representing the healthier or stronger pups that still show striking differences in GFR and proteinuria compared to normoxic offspring.

Another finding about which we are very grateful to Reviewer #2, for without this comment we would not have mentioned in our paper, are the striking differences we found in the postnatal growth curves of the offspring, including catch up growth (especially of hypoxic KO offspring) that further support the main message of our study (Fig. R2-2C, also new Fig. 1g).

In conclusion, according to Reviewer #2 these additional results should provide enough substantial evidence that **our adult studies stand on equal footing** and are thus very meaningful regarding the long term effects of fetal hypoxia on renal function. Below, the amended main text and the new images for Supplementary Figure 1 (Figs. R2-2A,B) and Figure 1 (Fig. R2-2C) can be found.

Amended main text: [...] Tracking the litter size of hypoxic and normoxic dams confirmed that the numbers of pups per litter were comparable and remained constant throughout the suckling phase (Supplementary Fig. 1d). Cannibalism of pups was an exception in our study. If it occurred, it usually affected the weakest pup in the afflicted litters in the first few days after birth, regardless of the experimental condition. Thus, no difference was evident in survival assessments (Fig. 1f). Importantly, we found striking differences in the postnatal growth of the offspring in relation to sex, genotype, and experimental condition, including a pronounced catch-up growth of hypoxic pups (Fig. 1g and Supplementary Figs. 1e,f). [...]

Fig. R2-2. Analysis of postnatal growth and survival of pups. (A) The number of pups per litter of normoxic and hypoxic dams did not significantly change in the first 4 weeks after birth. Multiple t tests. (B) The survival of pups between these 2 groups was not significantly altered. Log-rank (Mantel-Cox) or Gehan-Breslow-Wilcoxon tests. (C) Characterization of postnatal growth. At birth hypoxic offspring (grey lines) weighed significantly less than normoxic offspring (dotted lines), regardless of the genotype (small and large asterisks). Hypoxic KOs (bold grey line) remained lighter than normoxic KOs (bold dotted line) in the first week after birth (large asterisks), showed similar weights in the second postnatal week, but became significantly heavier during their

catch-up growth in the third week after birth (large asterisks), before the weight converged to a common level in week 4 and later. A similar trend, although less pronounced, was found between hypoxic and normoxic wt pups (small asterisks). Generally, the weight of KO pups was less compared to wt pups, showing significant differences until 8 weeks after birth for hypoxic offspring (grey circles), and during weeks 4 and 5 for normoxic offspring (white circle). Multiple t tests.

3. It would be important to take on EM studies with immunogold to demonstrate the changes with Fetuin A, and its localization. This is also important in the case of the KO mice employed in this study.

We appreciate Reviewer #2's idea to take on EM studies using immunogold, however we think it would just go beyond the scope of this paper, as Fig 3 already shows the detailed localization of Fetuin A, and we have added some supplementary stainings in our revised version. Moreover, immunogold labeling is a complex method, which can take up to several months to be properly established, and it was also not entirely clear to us, what exactly should be stained and where (Renal vs. hepatic Fetuin-A? Localization of Fetuin-A in normoxic or hypoxic animals or in biominerals?) Also, fetuin-A cannot be stained in *Ahsg* KO mice.

4. Figure 4, blots are simply shown. It is important to provide quantification of these blots as well.

The blots shown in Figure 4 clearly provided significant results. Therefore, we think addressing this point will not improve this study, but would rather generate redundant data that would unnecessarily expand the supplementary data file. Furthermore, we have moved Figs. 4d,e to Supplementary Figs. 6a,b to make room for new images obtained by answering point 7.

5. No power analysis is provided towards pre-determining the sample size for these studies. It appears that the sample size was arbitrarily chosen. This is a major weakness of this paper.

We thank Reviewer #2 for bringing up this comment. We agree that a power analysis greatly improves the integrity of a publication, provided that a study can rely on previous findings, where the variabilities among the study parameters are known and the effect of a treatment can be approximated beforehand. Our study on the other hand, is an explorative study based on the results of a gene expression analysis between normoxic and hypoxic fetal kidneys. Fetuin-A was the gene showing the highest induction in hypoxic kidneys and caught our undivided attention, given its prognostic importance for the progression of chronic kidney disease in humans. In the wake of this initial finding our goal was to investigate the mechanisms underlying the ectopic induction of this protein in the kidney and to clarify its role in hypoxia-induced IUGR and kidney development. Thus, a power analysis to pre-determine the sample size for our studies was not adequately possible. We decided to use a manageable amount of animals (6-7 per group, repeated in at least 3 three independent experiments) to examine the open questions. We hope these arguments explain why a power analysis was not provided. However, in follow-up studies that we are planning to address the therapeutic benefit of fetuin-A, the values obtained in this study will definitely serve as reference values in order to perform power analyses to calculate the appropriate sample sizes.

Retrospectively, the number of animals per groups was not arbitrarily chosen, but rather matched the minimum sample size requirements to detect differences with an alpha error = 0.05 and a beta error = 0.2, corresponding to a power of 80 %. For example, in our GFR analysis the variability of each group was around 10 % and the smallest difference between the groups was approximately 15 %, which in a power analysis results in a sample size of 7 animals per group. Similarly, for the proteinuria measurements the variability of each group was around 15 % and the smallest difference between the groups was approximately 25 %, which in a power analysis results in a sample size of 6 animals per group.

6. It was not clear if males or females were examined, since it is critically important to study both sexes, to determine if this phenomenon is sex-specific, or affects both.

Indeed, we had not clearly separated males and females in the first submission of the manuscript. We totally agree with Reviewer #2 that it is critically important to study both sexes. That's why we started a novel round of experiments with the goal to include 6-7 animals of each sex into our assessment of the long-term consequences of fetal hypoxia in wt and *Ahsg* KO mice. After a very time-consuming breeding period, we repeated our experiments and found similar results for both sexes regarding a reduced GFR or an enhanced proteinuria in KO and hypoxic animals. However, we also uncovered a sex-specific difference in the degree of

proteinuria, which was significantly higher in males than in females. No sex-specific difference was found for the GFR. Below, the amended main text and the new images for Figure 6 (Fig. R2-3) can be found.

Amended main text: [...] we measured urinary protein levels and determined the glomerular filtration rate (GFR) in adult female and male mice (Figs. 5a,b). GFR was reduced, whereas proteinuria was severely increased in both sexes of 9 weeks old *Ahsg* KO animals and mice exposed to fetal hypoxia compared normoxic controls. Whereas the values for GFR was comparable between the sexes, the degree of proteinuria was much higher in males than in females. Furthermore [...]

Fig. R2-3. Decline of renal function in adult hypoxic mice. (A) The increase in the protein/creatinine ratio was more pronounced in males than in females. For each sex, the ratio increased the most in hypoxic *Ahsg* KO animals, showing an additive effect of hypoxia and fetuin-A deficiency. (B) The decline in GFR was indistinguishable between the sexes. The greatest functional reduction was again present in hypoxic *Ahsg* KO animals.

7. While the discussion mentions other organs, it is important to have included another organ system in which such changes are seen or not seen, to determine if these observations are kidney specific, and not a generalized phenomenon in the hypoxia-induced IUGR offspring.

We believe that our model probably applies to other organs and might not be just specific for the kidney. Basis for this hypothesis is the observation that any kind of cellular stress, including chronic hypoxia, will negatively affect mitochondrial function, especially their ability to regulate calcium storage and transport [11]. A disturbance of this mitochondrial function will lead to mineral stress and fetuin-A might play an important role to protect the affected tissues from calcification and fibrotic remodeling. To gain insight whether other fetal organs showed signs of fibrotic stress under chronic hypoxic conditions, we determined the expression levels of several fibrotic markers in fetal hearts, kidneys and lungs. Indeed, we found that compared to the fetal kidney (Fig. R2-4A, also new Fig. 4d) also the fetal lung (which is also comprised of epithelial cells) showed an enhanced expression of collagens and fibronectin, in chronic hypoxia (Fig. R2-4B, also new Supplementary Fig. 6c). In contrast, the fetal heart seemed to be largely unaffected by chronic hypoxia as we observed no gross changes in gene expression (Fig. R2-4A, also new Supplementary Fig. 6d). Based on these findings, we conclude that epithelial organs are more likely to be affected by fetal hypoxia than non-epithelial organs such as the heart. Furthermore, different organ systems seem to have organ-specific response profiles, because not all fibrotic genes are regulated in a similar manner in hypoxia. For example, fetal hypoxic lungs had significantly reduced expression levels of *Acta2*, while this fibrotic marker was strongly induced in fetal hypoxic kidneys and also in the fetal heart. We hope that these preliminary results provide evidence that our findings seem to be a generalized phenomenon and is not restricted to the kidney. A more comprehensive analysis is surely needed to fully explain the effect of fetal hypoxia and the role of Fetuin-A in other organs, but would definitely go beyond the scope of this manuscript. We plan to address these open questions in follow-up publications. Below, the amended main text and the new images for Figure 4 (Fig. R2-4A) and Supplementary Figure 6 (Figs. R2-4B,C) can be found.

Amended main text: [...] Taken together, these findings identified fetuin-A as a novel, evolutionary conserved HIF-dependent target gene. Moreover, hypoxia not only promoted the expression of fetuin-A, but also stimulated the expression of several fibrotic marker genes in fetal kidneys (Fig. 4d), lungs and hearts (Supplementary Figs. 6c,d). However, whereas as the epithelial organs (lung and kidney) showed a broad activation of fibrotic genes, the response in the heart was more blunted and did not include an enhanced transcription of collagens.[...]

Fig. R2-4. Fibrotic response to hypoxia in other organs. (A) The fetal heart showed no changes in the expression of collagens or fibronectin in chronic hypoxic conditions. *Acta2* and vimentin showed a significant induction, although on a very low level. (B) The fetal kidney showed an induction of *Col3a1* and *Col6a1*, as well as of *Acta2* and vimentin in hypoxia. No change in gene expression was found for *Col1a1* or fibronectin. (C) The expression of fibrotic genes in hypoxic fetal lungs was much more similar to the fetal kidney than to the fetal heart. All assessed collagens and fibronectin were induced, while vimentin was not changed. Interestingly, *Acta2* was reduced in hypoxic lungs, while the expression of this fibrotic marker was enhanced in fetal hypoxic hearts and kidneys. Unpaired t tests. ****, $P < 0.0001$; ***, $P < 0.001$; **, $P < 0.01$; *, $P < 0.05$; ns, not significant.

Reviewer #3 (Remarks to the Author):

In their study Rudloff and colleagues using a valid murine model of fetal hypoxia-induced intra-uterine growth restriction (IUGR) at first instance characterized its short and long term consequences on the kidney. Importantly applying this hypoxic model in the *Ahsg* KO, i.e. fetuin-A KO mice revealed the highly interesting finding that fetuin-A, being a well-known calcification inhibitor, also acts as an evolutionary conserved HIF (hypoxia-inducible transcription factor) target gene, by which it protects IUGR kidneys from fast progression to CKD through clearance of calcium particles thereby attenuation microphage infiltration, inflammation and fibrosis.

This is an interesting study shedding new light on fetuin-A function which on the longer run might also have therapeutic potential not only in the field of CKD but also in other hypoxia-mediated disease states. The study set-up of the present study is straightforward and very well outlined with various *in vivo* as well as *in vitro* studies following up each other in a logic way, thereby confirming and proving evidence to each other. The authors propose a nice mechanistic model (Figure 8). Although the proposed model sounds convincing, going through the paper, this reviewer leaves a bit with a 'the chicken and the egg' story in that one may ask the question if fetuin (i) directly protects the kidney against hypoxia-induced renal damage thereby inhibiting ectopic calcification or (ii) through inhibition of calcium mineral deposition/maturation rescues the kidney from damage and ensuing CKD. I would not see this comment as a negative point, however, would propose to deal herewith in the discussion section of the paper.

We want to thank Reviewer #3 for the constructive feedback. Regarding "the chicken and the egg" story we want to point out that the main physiological function of fetuin-A is the inhibition of ectopic calcium mineral maturation and deposition, which generally protects the body from organ damage and pathophysiological fibrotic remodeling of calcified tissue and accompanying diseases [12]. According to our findings, this physiological role of fetuin-A can be locally enhanced in the kidney by the activation of ectopic fetuin-A production during hypoxia, which provides a boost mechanism to augment the capacity to clear the increased release of calcium minerals during hypoxic cellular stress. This enhanced clearing power together with the suppression of the inflammatory response, in turn protects the kidney from further mineral-stress induced renal damage. Furthermore, we do not think that fetuin-A directly protects against hypoxia-induced damage, but rather is a key player in safeguarding the integrity of the tissue by keeping the damage in check. For a clearer conveyance of our proposed model, we have amended the text in the discussion as follows:

Amended main text: [...] we identified *Ahsg* as a novel hypoxia target gene that locally protects IUGR kidneys from chronic, progressive renal damage induced by prenatal hypoxia. **In Figure 8, we propose a model in which the systemic function of liver-derived fetuin-A (green circles) can be locally enhanced upon hypoxic cellular stress. This locally produced, renal fetuin-A (yellow circles) provides a boost mechanism that augments the capacity in the kidney to clear the increased release of calcium minerals from stressed cells and to suppress the inflammatory response. This in turn protects the kidney from further mineral-stress by keeping the renal damage in check. An inability to activate the local fetuin-A production otherwise stimulates macrophage infiltration, inflammation and TGF- β mediated fibrotic tissue remodeling, leading in the long-term to proteinuria and decreased GFR in the adult. [...]**

In line herewith, one should put the potential effect (potential therapeutic use) of fetuin A on hypoxia-induced renal damage (perhaps other organs) in the setting of e.g. ischemia/reperfusion acute kidney injury in which, to the best of this reviewer's knowledge no increased calcium deposition has been demonstrated so far, in perspective.

Indeed, we have begun to address a potential therapeutic role of fetuin-A in various renal pathologies and have very recently filed a European patent application, which we have mentioned in the revised version of the paper. This question however will be addressed in at least one if not more follow-up manuscripts. In a first set of preliminary experiments, we found that small-scale calcium-containing biominerals are actually present in at least 2 other mouse models of kidney injury (Unilateral Ureter Obstruction UUO and Ischemia Reperfusion Injury IRI). To detect these particles we made use of 488-fetuin-A and its high binding specificity for calcium and staining sensitivity (Figs. R3-1). Furthermore, we could also already show that intraperitoneal injection of fetuin-A reduced the expression of fibrotic markers in these injury models (Figs. R3-2). We believe that these novel findings will path the way to a potential therapeutic or preventive application of Fetuin-A in IRI.

Fig. R3-1. Deposition of calcium biominerals in renal injury mouse models as detected by staining with 488-Alexa-labelled fetuin-A. (A) Ischemia reperfusion injury (IRI) was induced in the left kidney by clamping the renal vessels for 30 min, the right kidney served as a control. 7 days post-surgery, the kidneys were isolated and analyzed. Minute calcium particles (white arrowheads) were present in kidneys that underwent 30 minutes warm ischemia, but absent in controls. (B) Unilateral ureter obstruction (UUO) was performed by double-ligating the left ureter according to standard protocols, the right kidney served as a control. 3 days post-surgery, the kidneys were isolated and analyzed. Small-scale calcium particles (white arrowhead) were also detectable in UUO kidneys, but not in controls.

Fig. R3-2. Therapeutic effect of fetuin-A in an IRI mouse model. Immediately after IRI surgery, mice received the first treatment of either physiological NaCl solution or bovine Fetuin-A monomer (100µg/g body weight) via intraperitoneal injection. Injections were repeated daily for 4 days. Kidneys were isolated on day 5 and analyzed. The expression levels of *Col1a1* (A), *Col3a1* (B), and *Col6a1* (C) were significantly enhanced in kidneys subjected to IRI. This induction could be significantly reduced upon treatment with fetuin-A.

Some minor comments:

Figures, Graphs: Data in figures alternately are presented as either individual values, box-whisker plots, average with SD bars. Perhaps it might be wise to present data in a way that to a certain extent covers the various ways of presentation; i.e. individual data presented as dot plots next to the average for the group with appropriate error bars.

We changed the presentation of the data to a uniform way, showing in individual data points.

The number of animals included for the various experiments/analyses is not always well defined; e.g. ≥ 5 or ≥ 4 . I would propose that the exact number should be mentioned.

Since we change the presentation of our data (see previous comment), the number of animals can be directly derived from the graphs.

Introduction:

Last para: In case additional data on the characterization of the hypoxic model as well as the KO mice models are available in literature or unpublished, reference should be made to these in e.g. the reference list or supplemental material.

References to hypoxic mouse models and the KO mouse lines used are given when these are introduced in the text and can be found in the reference list.

Results:

Maternal weight gain was significantly reduced in hypoxic dams going along with a decreased LBW and number of nephrons in the hypoxic fetuses, it might be worth the effort to calculate the LBW/maternal weight and number of nephrons/LBW ratio's in the various groups and include these data in the supplementary information.

This is an interesting point and we calculated the E18.5 fetal weight/maternal weight ratio for normoxic (0.0308), hypoxic (0.0303) and Cc mothers (0.0302) and found no statistically significant difference among the groups (Fig. R3-3, also new Supplementary Fig. 1c). Unfortunately, we could not perform a similar calculation for the ratio of the number of nephrons/E18.5 fetal weight, because we could not allocate the iDisco-stained kidneys to the weight of the embryos, and we did not perform this staining for Cc fetal kidneys. Nonetheless, using the average values we determined that 1411 nephrons were formed per gram in normoxic embryos, and only 1300 per gram in hypoxic LBW embryos. This corresponds to a reduction of 7.9 % in hypoxic embryos. Given that the difference in the E18.5 fetal weight/maternal ratio is neglectable, we assume that the difference in the number of nephrons/E18.5 fetal weight might be significant. These results show that 1) the weight of the fetuses correlates with maternal weight, and 2) that the formation of nephrons is indeed severely compromised in small for gestational age fetuses exposed to chronic hypoxia. The findings of our study now provide a mechanism for why this happens. We incorporated these results into our study as follows:

Amended main text: [...] whereas normoxic and Cc fetuses did not (Fig. 1d). **Calculating the E18.5 fetal weight/maternal weight ratio revealed no significant difference among the groups (Supplementary Fig. 1c), which showed that fetal weight seems to be dependent on maternal weight.** Kidneys of hypoxic fetuses were smaller with significantly fewer nephrons compared to controls (Fig. 1e, Supplementary Fig. 2 and Supplementary Videos 1,2). **Moreover, the number of nephrons/E18.5 fetal weight ratio was 1411 in normoxic vs. 1300 in hypoxic fetuses, which further illustrates that nephrogenesis was severely disturbed in chronic hypoxia and does not correlate with fetal weight.** [...]

Fig. R3-3. E18.5 fetal weight/maternal weight ratio. No difference among the experiment groups was detected.

There was a reduction in the number of nephrons. Did the authors also have the possibility to check the intactness of the nephrons; e.g. was there a difference in the incidence of e.g. atubular glomeruli.

We did not observe morphological abnormalities concerning the intactness of nephrons.

Page 7: lines 131 - 135: '... fetuin was expressed only in the PT of hypoxic fetal kidneys ...'. This cannot be derived from figure 3i-l as data from normotoxic kidneys are not presented.

All images in Figure 3 have to be viewed in a mutual context that fetuin-A is produced in the PT of hypoxic kidneys. The expression pattern of fetuin-A in normoxic kidneys is shown in Figs. 3a-d. The images in Figures 3i-l serve to visualize the colocalization of fetuin-A with different nephron segment markers. Here the signal of

fetuin-A only overlaps with that of aquaporin 1, which marks the PT. We kindly ask Reviewer #3 to reconsider his comment.

Page 9, line 191: 189 – 192: ‘... positive 488-FA staining in the absence of von Kossa or Alizarin-Red ... structures merely enriched with calcium ... calcium-phosphorus aggregates ...’ Has this statement been evidenced in literature. If so reference should be made to it.

Ectopic fetuin-A has been used to stain calcium aggregates in *Ahsg* KO mice in a recent publication by Herrmann et al. assessing the cause of morbidity in *Ahsg* KO mice [13]. Furthermore, 488-FA has been used to generate CPPs to follow their trajectories in vivo [14]. These references have been added to the literature.

Page 9, line 195: this sentence sounds rather speculative.

We have removed the second part of this sentence to avoid a speculative tone.

Page 9, line 195, figures 6b,c: No staining is shown for the normotoxic *Ahsg* KO’s. The same counts for Figure 7. Was there any reason for this.

We showed the staining for normoxic *Ahsg* KO kidneys in Figure 6k. Except for the absence of fetuin-A (red), no striking differences can be detected in direct comparison with normoxic wt kidneys shown in Figure 6j. The images in the left panel of Figure 6 primarily served to demonstrate the deposition of small-scale calcium-containing biominerals that can only be found in hypoxic KO kidneys (see also the diffuse green staining in Figure 6m). In the revised manuscript Figure 6 was moved to Figure 5.

Unfortunately, we could not follow the remark regarding Figure 7, because we do not understand what Reviewer #3 refers to.

Page 10, line 214: As effects on *Tgfb1* are an essential part in the reasoning I would propose to include the *Tgfb1* figure in the figures of manuscript rather than including it in the supplementary information

We followed the advice of Reviewer #3 and moved the *TGFB1* data into main Figure 7.

Page 10, lines 215 -218: I would propose to turn around this sentences; i.e. ‘... fibrotic markers were generally enhanced in hypoxic culture conditions and further increased in *Ahsg* KO pPTCs ...’

We followed the advice of Reviewer #3 and turned this sentence around.

Page 210, line 224: Could we, as an alternative name this a ‘soluble decoy receptor protein’ also.

We have added this description to the text.

Discussion:

Page 12, 251-251: In order to put the model in a clinical perspective it might be of interest to translate the degree of renal insufficiency in the mouse model to a certain CKD .. stage in humans.

The suggestion to translate our results into a clinical perspective is a highly stimulating evaluation of our results and gives our findings an even better, more valid interpretation basis. We used the KDIGO 2012 staging classification guidelines for GFR and proteinuria to categorize the risk of our experimental animals for CKD (Fig. R3-4, also new Supplementary Fig. 10). Since already healthy mice exhibit much higher urinary protein levels than humans (males showing even higher values than females), we introduced 2 elevated, sex-specific threshold levels for this staging category based on the results of our wt mice. The steps indicative of worsening of the disease were kept similar to the human situation. For the staging of GFR, we defined a GFR of >900 ($\mu\text{l}/\text{min}/100\text{g BW}$) as healthy and kept the grading steps similar to the human situation. Based on these assumptions, we found that all animals except normoxic wt mice (males and females) had an increased risk for CKD; all remaining males even showed a high risk for CKD. We implemented this translational aspect into the discussion of our results as follows:

Amended main text: [...] where late gestational hypoxia alone was sufficient to reduce GFR and to induce proteinuria in adult IUGR mice. **If these findings were to be translated into a clinical perspective, then hypoxic or *Ahsg* KO animals would have a moderate to high risk for CKD (Supplementary Fig. 10), which reflects the clinical course of CKD in human subjects born as IUGR babies. [...]**

Translation of renal insufficiency in our IUGR mouse model to CKD risk groups in human				Proteinuria categories (mg/mmol Cr)					
				Description and sex-specific ranges					
				P1		P2		P3	
				Normal to mild increase		Mild to moderate increase		Severe increase	
F: <90 M: <190		F: 90-120 M: 190-220		F: >120 M: >220					
GFR categories (µl/min/100g BW) Description and range	G1	Normal to high	>900	No wt F	No wt M	No KO F	Hy wt F	Hy wt M	No KO M
	G2	Mild decrease	600-899					Hy KO F	Hy KO M
	G3a	Mild to moderate decrease	450-599						
	G3b	Moderate to severe decrease	300-449						
	G4	Severe decrease	150-299						
	G5	Kidney failure	<150						
KDIGO 2012 risk classification		low risk		moderate risk		high risk		very high risk	

Fig. R3-4. Translation of our IUGR mouse model results into a clinical perspective, using KDIGO 2012 classification of CKD as a basis. Mice data are bold. Wt mice (males and females) have no or only a low risk for CKD. Normoxic *Ahsg* KO females and hypoxic wt females have a moderate risk for disease, whereas all remaining males and hypoxic *Ahsg* KO females are staged into a high risk category. F, female; M, male; No, normoxic; Hy, hypoxic; wt, wild type; KO, *Ahsg* KO.

Methods:

Page 16: Was there any mortality in the animals?

We did not observe mortality in the animals during the experimental time frame.

Page 19, RT-qPCR: Which housekeeping gene was used?

We used *Actb* (adult animals) or *Ppia* (fetuses) as housekeeping genes.

Patrick C. D'Haese

Literature

1. Chatterjee, P., et al., *Adipocyte fetuin-A contributes to macrophage migration into adipose tissue and polarization of macrophages*. J Biol Chem, 2013. **288**(39): p. 28324-30.
2. Eltzschig, H.K. and P. Carmeliet, *Hypoxia and inflammation*. N Engl J Med, 2011. **364**(7): p. 656-65.
3. Jahnen-Dechent, W., et al., *Fetuin-A regulation of calcified matrix metabolism*. Circ Res, 2011. **108**(12): p. 1494-509.
4. Vuguin, P.M., *Animal models for small for gestational age and fetal programming of adult disease*. Horm Res, 2007. **68**(3): p. 113-23.
5. Ergaz, Z., M. Avgil, and A. Ornoy, *Intrauterine growth restriction-etiology and consequences: what do we know about the human situation and experimental animal models?* Reprod Toxicol, 2005. **20**(3): p. 301-22.
6. Girard, J.R., et al., *Fetal metabolic response to maternal fasting in the rat*. Am J Physiol, 1977. **232**(5): p. E456-63.
7. van de Looij, Y., et al., *Nutritional Intervention for Developmental Brain Damage: Effects of Lactoferrin Supplementation in Hypocaloric Induced Intrauterine Growth Restriction Rat Pups*. Front Endocrinol (Lausanne), 2019. **10**: p. 46.
8. Jimenez-Chillaron, J.C., et al., *Beta-cell secretory dysfunction in the pathogenesis of low birth weight-associated diabetes: a murine model*. Diabetes, 2005. **54**(3): p. 702-11.
9. Garofano, A., P. Czernichow, and B. Bréant, *Postnatal somatic growth and insulin contents in moderate or severe intrauterine growth retardation in the rat*. Biol Neonate, 1998. **73**(2): p. 89-98.
10. LUBCHENCO, L.O., et al., *INTRAUTERINE GROWTH AS ESTIMATED FROM LIVEBORN BIRTH-WEIGHT DATA AT 24 TO 42 WEEKS OF GESTATION*. Pediatrics, 1963. **32**: p. 793-800.
11. Boonrungsiman, S., et al., *The role of intracellular calcium phosphate in osteoblast-mediated bone apatite formation*. Proc Natl Acad Sci U S A, 2012. **109**(35): p. 14170-5.
12. Schafer, C., et al., *The serum protein alpha 2-Heremans-Schmid glycoprotein/fetuin-A is a systemically acting inhibitor of ectopic calcification*. J Clin Invest, 2003. **112**(3): p. 357-66.
13. Herrmann, M., et al., *Luminal calcification and microvasculopathy in fetuin-A-deficient mice lead to multiple organ morbidity*. PLoS One, 2020. **15**(2): p. e0228503.
14. Köppert, S., et al., *Cellular Clearance and Biological Activity of Calciprotein Particles Depend on Their Maturation State and Crystallinity*. Front Immunol, 2018. **9**: p. 1991.

REVIEWER COMMENTS

Reviewer #1 (Remarks to the Author):

Comments on the revised Ms entitled "Fetuin-A, a novel HIF target, safeguards tissue integrity during hypoxic stress" (NCOMMS-19-15730).

On going through the abovementioned revised Ms, it has been found that the authors have modified the Ms by a few additional experiments and altered the text accordingly. They have also provided a detailed rebuttal on the reviewers comments. However, some of the aspects of their study still remain to be addressed. I am providing below a few comments on these which require authors attention.

1. Figure 2F is removed in the revised Ms. The point that we mentioned about this figure was about the serum Fetuin-A (FetA) level of the fetus. We also requested the authors to provide the serum FetA value of mother during hypoxic and normoxic conditions. However, the authors this time provided these values in mother and fetus in the rebuttle in two different places but not in the text. According to them there is no change of serum FetA in mother and fetus during hypoxic condition. However, it is not clear from the FetA ELISA value in mother. These values in fetus and mother are important and we would suggest them to include them in the text. ELISA data provided are not very clear in the figures and those should to be more clearer. Since, further experiments showed a significant difference between mother and fetus, abovementioned data are therefore relevant.

2. It is shown by the authors that hypoxia induces FetA gene expression in the liver and kidney of the fetus, but serum level of FetA in the fetus remain the same. According to the authors this has local, rather than systemic functional relevance. But, being a secretory protein (Trepanowski et al., Nature, 2015; Brix et al., J Clin Metab, 2010; Jensen et al., Diabetes care, 2013; Stefan et al., Diabetes, 2008), it is surprising how the serum FetA concentration in fetus remains unaltered even when there is an increase of FetA in the kidney and liver.

3. Increase in FetA has been shown to be related to the increase in fat/lipid by several authors (Stefan et al., Nature medicine, 2013; Trepanowski et al., Nature, 2015; Haukeland et al., European journal of Endocrinology, 2013). In fetus, increase is occurring most likely without the presence of excess lipid. The authors have shown that FetA elevation is effected by HIF which is mentioned as one of its transcription factor and HIF is induced by hypoxia. Hence, FetA elevation awaits the increase in HIF. The sequence of events in kidney therefore appears to be hypoxia-HIF-FetA, therefore, FetA increase is much later as compared with the initiation of hypoxia induction. Therefore, question is whether FetA protects the damage or it repairs the damage? From the sequence of events mentioned above it seems that FetA repairs the damage but authors described FetA's protective role.

4. The hypoxic condition has not been able to induce FetA gene expression in the liver and kidney of mother. But, HIF, as a transcription factor, may also act in the mother to increase the FetA gene expression, but the authors demonstrated that there was no change in FetA expression in mother. What could be the explanation for it? If hypoxia induces HIF in fetus kidney resulting tissue damage which according to author is prevented by FetA, but same was occurring in mother without any tissue damage. It would be better if the reason behind such differences could be explained. Interestingly, circulatory level of FetA doesnot alter in neither cases i.e. in mother and fetus. Questions like these were also raised earlier but authors have not included anything in this direction in the revised Ms.

5. Please note that only the infiltration of macrophage (F4/80) to a tissue does not indicate inflammation because circulatory macrophage usually remains as M2 (alternatively activated) and these are anti-inflammatory, whereas entering into the tissue is a shift of phenotypic polarization occurs from M2 to M1 (classically activated) which are pro-inflammatory in nature and that causes inflammation. Hence, the authors require to determine how much of infiltrated macrophage is converted from M2 to M1 in the fetus of kidney to ascertain macrophage induced inflammation in hypoxic fetus kidney.

Reviewer #3 (Remarks to the Author):

The authors very well addressed my comments made at first revision.

Reviewer #4 (Remarks to the Author):

The authors use a well-established murine model of hypoxia-induced intrauterine growth restriction (IUGR) to characterize the effect of impaired fetal growth on the kidney. Using a fetuin-A (Ahsg) KO model, the experiments described demonstrate that fetuin-A is an HIF target gene and protects against CDK in IUGR cases.

For clarity to the authors: Given that I did not review the first submission of this paper, I focused my attention on the responsiveness of the authors to initial concerns raised by the reviewers and read through the manuscript to identify any additional issues. In my view, the authors were extremely thorough in their responsiveness to reviewer concerns and made substantial improvements to their manuscript. One of my predominant concerns would have been the effect of hypoxia to reduce maternal caloric intake on the primary outcomes of interest, and that was sufficiently addressed by the authors.

A few other comparatively minor issues noted:

Line 40: LBW is traditionally defined as a birth weight < 2500g rather than < the 10th percentile. LBW is not always a consequence of IUGR (i.e., there are lots of reasons an infant can be born LBW, including non-pathological causes).

Line 57: "Population studies conducted among highland dwellers of various ethnicities found a clear correlation between altitude and incidence of renal disease¹¹ making high altitude pregnancy the most important cause of maternofetal hypoxia and IUGR worldwide" The connection between the two sections of this sentence is unclear. (i.e., Why does a correlation between altitude and renal disease make HA pregnancy the most important cause of maternofetal hypoxia?)

Line 61: Use of the word proof is too strong. Suggest "robust mechanistic evidence to support Barker's hypothesis" or something similar.

Did authors consider fetal:placental ratio as an index of placental insufficiency?

Line 111: This is the first mention that a KO model was used and therefore seems to pop out of nowhere. The authors should include some information in the abstract and introduction re: the KO model used for the experiments. I found both the abstract and the introduction to be too lean, lacking detail about experiments/models (abstract) and background (introduction).

Point-to-point rebuttal to the Reviewers' comments

We would like to thank all reviewers again for their helpful comments. We have addressed them in the 2nd revision of our manuscript and believe that they greatly enhanced the quality of our study. In the following passages, our response to each reviewer is giving point by point.

Reviewer #1 (Remarks to the Author):

Comments on the revised Ms entitled “Fetuin-A, a novel HIF target, safeguards tissue integrity during hypoxic stress” (NCOMMS-19-15730).

On going through the abovementioned revised Ms, it has been found that the authors have modified the Ms by a few additional experiments and altered the text accordingly. They have also provided a detailed rebuttal on the reviewers comments. However, some of the aspects of their study still remain to be addressed. I am providing below a few comments on these which require authors attention.

1. Figure 2F is removed in the revised Ms. The point that we mentioned about this figure was about the serum Fetuin-A (FetA) level of the fetus. We also requested the authors to provide the serum FetA value of mother during hypoxic and normoxic conditions. However, the authors this time provided these values in mother and fetus in the rebuttle in two different places but not in the text. According to them there is no change of serum FetA in mother and fetus during hypoxic condition. However, it is not clear from the FetA ELISA value in mother. These values in fetus and mother are important and we would suggest them to include them in the text. ELISA data provided are not very clear in the figures and those should to be more clearer. Since, further experiments showed a significant difference between mother and fetus, abovementioned data are therefore relevant.

Reviewer #1 wrote “Figure 2F is removed in the revised Ms”. We would like to point out that this is not correct: Figure 2f was not removed from the 1st revision of the manuscript, but replaced by Figure 2e, where we repeated the assessment of serum fetuin-A levels of E18.5 fetuses (comprising a much larger sample size than before) using the Mouse Fetuin A/AHSG Quantikine ELISA Kit (R&D, MFTA00). Along with the measurement of fetal fetuin-A levels in the serum, we also determined maternal serum fetuin-A levels. This data was shown in Supplemental Figure 3f of the 1st revision of the manuscript. Furthermore, these findings were already also mentioned in the text in last sentence of the 2nd paragraph of the 1st revision. We admit that the figure labelling of the maternal data was not the same in the rebuttal and the 1st revision of the manuscript, and we apologize for this inconsistency.

To improve this point of criticism raised by Reviewer #1, we repeated the measurements and now show the ELISA data of fetuses and mothers side by side in Figures 2d and 2e, respectively, in the 2nd revision of the manuscript. We did not find significant differences in both data sets. Furthermore, we also included the values in the text. Regarding the presentation of the ELISA data, we adhere to the reproducibility guidelines of Nature Communications from 2018 [1], and had shown all data points for the ELISA measurements already in the 1st revision, which in our opinion is the best way to present the data.

The text of the result section was amended as follows:

Despite its strong induction in hypoxic kidneys, we did not detect a significant rise of **serum fetuin-A levels in hypoxic fetuses ($68.7 \pm 4.3 \mu\text{g/ml}$) compared to normoxic controls ($70.5 \pm 3.8 \mu\text{g/ml}$) (Fig. 2d), nor in their mothers (Fig. 2e: normoxia: $190.0 \pm 14.1 \mu\text{g/ml}$ vs. hypoxia: $172.1 \pm 14.8 \mu\text{g/ml}$).**

Figures 2d,e

2. It is shown by the authors that hypoxia induces FetA gene expression in the liver and kidney of the fetus, but serum level of FetA in the fetus remain the same. According to the authors this has local, rather than systemic functional relevance. But, being a secretory protein (Trepanowski et al., Nature, 2015; Brix et al., J Clin Metab, 2010; Jensen et al., Diabetes care, 2013; Stefan et al., Diabetes, 2008), it is surprising how the serum FetA concentration in fetus remains unaltered even when there is an increase of FetA in the kidney and liver.

We never showed that hypoxia enhanced Fetuin-A expression in the fetal liver. In fact, no gene that showed an induction in the fetal kidney upon hypoxia actually also showed increased transcription in the fetal liver. However, that the liver is able to respond to fetal hypoxia is nicely exemplified by the induction of Epo (Supplementary Fig. 3a). Furthermore, we would like to draw your attention to the fact that the expression levels depicted in Figs. 2b,c and Supplementary Figs. 3b-f (renal – left, hepatic – right) are shown on a logarithmic scale at the Y-axis. We chose this representation because the hepatic expression levels are several magnitudes higher than the renal expression levels (e.g. fetuin-A transcription is 600-fold higher) and that the expression of the analyzed genes is only induced in hypoxia and only in the kidney. To underline this important point, we explicitly mention the logarithmic nature of the Y-scale in the text and figure legends of the 2nd revision of the manuscript.

According to the data, there are 2 reasons, why fetal hypoxia does not affect the levels of fetuin-A in the serum. First, hypoxia does not provoke an increase in hepatic transcription of fetuin-A: since the liver is the main source of fetuin-A, no change in serum concentration is expected. Secondly, the amount of fetuin-A that becomes ectopically expressed in the proximal tubules of fetal hypoxic kidneys is 60-fold lower than the bulk protein mass secreted by the liver; thus, even if renal fetuin-A were secreted into the systemic circulation, its contribution to the systemic pool of fetuin-A would be minimal. Hence, renal fetuin-A cannot contribute to an increase in fetuin-A serum levels.

The developing kidney is one of the most vulnerable organs to hypoxia, and the ectopic expression of fetuin-A provides an additional protective mechanisms to safeguard nephrogenesis in times of increased hypoxic stress. Taken together our findings strongly suggest that fetuin-A with renal origin has a local, protective rather than a systemic effect. To emphasize these points, we included following text:

This is not surprising as under normal conditions the liver is the main source of circulating fetuin-A (600-fold higher than in the fetal kidney), whereas in our model, fetal hypoxia induced local, “ectopic” fetuin-A expression in the kidney, at a level which is still 60-fold lower than in the liver (Fig. 2b: note that the expression levels of fetuin-A in kidney and liver are shown on a logarithmic scale). These findings provide evidence that the induction of fetuin-A in hypoxic fetal kidneys does not have a systemic functional relevance, but rather a local, important protective role in the developing kidney.

3. Increase in FetA has been shown to be related to the increase in fat/lipid by several authors (Stefan et al., Nature medicine, 2013; Trepanowski et al., Nature, 2015; Haukeland et al., European journal of Endocrinology, 2013). In fetus, increase is occurring most likely without the presence of excess lipid. The authors have shown that FetA elevation is effected by HIF which is mentioned as one of its transcription factor and HIF is induced by hypoxia. Hence, FetA elevation awaits the increase in HIF. The sequence of events in kidney therefore appears to be hypoxia-HIF-FetA, therefore, FetA increase is much later as compared with the initiation of hypoxia induction. Therefore, question is whether FetA protects the damage or it repairs the damage? From the sequence of events mentioned above it seems that FetA repairs the damage but authors described FetA's protective role.

Of course, the induction of fetuin-A requires some time and thus it cannot protect the kidney from initial damage, but once produced it has a clear protective role in the developing kidney. We have already addressed this question in the 1st revision, which was similarly raised by Reviewer #3 and made appropriate changes in the manuscript.

In our opinion Fetuin-A fulfills both roles. Initially, fetuin-A has a damage mending role by clearing calcium biominerals, i.e. the trigger of tissue remodeling and inflammation. Once this initial trigger is removed, fetuin-A also fulfills a protective function by breaking the vicious circle of calcium biomineral-induced inflammation and tissue damage, thus reducing further damage and safeguarding renal tissue integrity.

To add further evidence, we demonstrated a protective effect of fetuin-A in an adult mouse model of hypoxia-related kidney damage. Similar to our findings in fetal hypoxia, we showed in adult kidneys undergoing ischemia reperfusion injury (IRI) an accumulation of calcium biominerals, using the sensitive 488-FA staining approach (novel Figure 8j,k). This is to our knowledge the first time that these very tiny calcium biomineral particles could be visualized in IRI kidneys, and we suggest that their formation is one of the hallmarks of ischemic tissue damage. Furthermore, supplementation of exogenous fetuin-A immediately starting after reperfusion reduced the expression of fibrotic markers in the injured kidney 5 days after surgery (novel Figure 8l,m). Thus, our study clearly shows that fetuin-A has protective functions in 2 independent mouse models of hypoxia-related tissue damage.

Moreover, the protective effect of fetuin-A goes beyond simple hypoxia and is even effective in reperfusion injury.

Following new paragraph **Fetuin-A supplementation reduces the expression of fibrotic markers upon ischemia reperfusion injury** was added to the manuscript:

In the previous sections we have presented results that describe fetuin-A as an important player in Barker's hypothesis, counteracting multiple disadvantageous processes in the fetal kidney. Because of the underlying pathophysiological mechanisms we anticipated that the protective role of fetuin-A is not restricted to the fetus, but can be extended to offset similar harmful processes in hypoxia-related injury in adult animals. Thus, in a final step, we performed an interventional study using a mouse model of ischemia reperfusion injury (IRI)⁴⁵. Here, renal blood flow is transiently stopped to induce hypoxic damage in the kidney (ischemia), which is further exacerbated upon the restoration of renal circulation (reperfusion). Similar to our fetal model, we showed for the first time that tissue damage in IRI kidneys was associated with the deposition of calcium containing microparticles (Fig. 8k), which were not found in controls (Fig. 8j). The presence of these deposits in IRI kidneys not only validated the superiority of the 488-FA staining approach to detect early calcium biominerals, but also corroborated our findings in fetal hypoxia, revealing an imbalance between calcium mineral release and clearance upon hypoxic tissue damage. Furthermore, daily administration of fetuin-A for 4 days, starting immediately after IRI surgery, resulted in a marked decrease of *Col1a1* and *Col3a1* compared to mice treated with physiological saline solution (Figs. 8l,m). In this regard it was reported in rats that peripheral fetuin-A administration could prevent excessive cerebral ischemic tissue injury⁴⁶. Our results provide strong evidence that fetuin-A supplementation at the time of injury (e.g. ischemia reperfusion injury in diverse organ systems) could be a promising novel therapeutic approach against hypoxia-induced mineral stress and fibrotic tissue remodeling, particularly in conditions associated with fetuin-A depletion, such as CKD or acute inflammation.

Novel Figure 8.

4. The hypoxic condition has not been able to induce FetA gene expression in the liver and kidney of mother. But, HIF, as a transcription factor, may also act in the mother to increase the FetA gene expression, but the authors demonstrated that there was no change in FetA expression in mother. What could be the explanation for it? If hypoxia induces HIF in fetus kidney resulting tissue damage which according to author is prevented by FetA, but same was occurring in mother without any tissue damage. It would be better if the reason behind such differences could be explained. Interestingly, circulatory level of FetA doesnot alter in neither cases i.e. in mother and fetus. Questions like these were also raised earlier but authors have not included anything in this direction in the revised Ms.

Exposure to 10 % oxygen for 4 days might not be enough to provoke obvious tissue damage in adult kidneys or other organs. Many studies make use of hypoxia protocols similar to ours and do not report on adult tissue damage [2-5]. Yet, the fetus and especially the fetal kidneys respond differently to maternal hypoxia, since the fetus is already exposed to borderline hypoxic conditions in normal physiological conditions [6, 7]. A major reason for the different observation between fetuses and mothers might be the fetal circulation. Gas exchange in the fetus occurs in the placenta instead of the lungs as in newborns or adults. Furthermore, the most highly oxygenated blood in the fetus (80-90 % saturated, coming from the placenta) is delivered to the brain and myocardium due to extra- and intracardiac circulatory shunt mechanisms. Hence, the oxygenation of more peripheral organs including the developing kidney is severely reduced in the fetus compared to the mothers (already at normal conditions). Thus, in the hypoxic group, the fetal kidney is exposed to much more stringent hypoxic conditions than the adult kidney. The fetal kidneys then respond to this extreme hypoxic environment with the induction of fetuin-A, which provides an additional safeguard mechanism to protect the developing kidney from these harsh hypoxic conditions. On the other hand, the degree of hypoxia in the maternal kidney is still not sufficient to induce the transcription of fetuin-A, even under hypoxic conditions. Lastly, as already mentioned above, the level of circulatory fetuin-A is mainly influenced by its release from the liver, and the liver is obviously producing fetuin-A independently of oxygen levels. This is due to the use of strong HIF independent promoter/enhancer elements.

In conclusion, we agree that extensive discussion of maternal/ fetal differences is a very interesting point, but believe that it does not significantly enhance the message of our study, which is focused on the fetal response to hypoxia. Nevertheless, to acknowledge this point, we included following section into the discussion:

A reason for the specific induction of fetuin-A expression in fetal hypoxic kidneys might be the fetal circulation with its extra- and intracardiac shunt mechanisms, delivering the most highly oxygenated blood to the brain and myocardium. This further reduces the oxygenation of more peripheral organs including the developing kidney already at normoxic conditions. Thus, in the hypoxic group, the fetal kidney is exposed to much more stringent hypoxic conditions, which then respond to this extreme hypoxic environment with the induction of fetuin-A.

5. Please note that only the infiltration of macrophage (F4/80) to a tissue doesnot indicate inflammation because circulatory macrophage usually remains as M2 (alternatively activated) and

these are anti-inflammatory, whereas entering into the tissue if shift of phenotypic polarization occurs from M2 to M1 (classically activated) which are pro-inflammatory in nature and that causes inflammation. Hence, the authors require to determine how much of infiltrated macrophage is converted from M2 to M1 in the fetus of kidney to ascertain macrophage induced inflammation in hypoxic fetus kidney.

We thank Reviewer #1 for bringing up this argument once more, since we did not thoroughly address this comment in the 1st revision of the manuscript. Now, we have used FACS to sort M1 and M2 macrophages of E18.5 fetal kidneys and our results show that 1) hypoxia profoundly disturbs the equilibrium of resident (F4/80^{hi}CD11b^{low}) and infiltrating macrophages (F4/80^{low}CD11b^{hi}) in the developing kidney [8, 9] and 2) shifts the polarization of kidney macrophages towards M1 (CD206⁻). 3) fetuin-A attenuates the number of infiltrating macrophages and mitigates the shift of polarization from M2 to M1. The switch from M2 to M1 might also be influenced by the low availability of oxygen during fetal hypoxia. Here, it was shown that M1 macrophages rely on glycolysis to obtain energy, whereas M2 macrophages depend on oxidative metabolic processes [10]. A reduction in tissue resident F4/80^{hi}CD206⁺ M2 macrophages might also have a diminishing effect on nephrogenesis, since it was shown that these cells have a trophic function in the developing kidney [11, 12]. Taken together, we show that hypoxia leads to an environment in the fetal kidney that promotes the infiltration and polarization of pro-inflammatory M1 macrophages at the cost of pro-proliferative M2 macrophages. That this phenotype is further aggravated in fetuin-A KO mice provides more evidence that fetuin-A has protective effects in hypoxic conditions.

The following new paragraph **Renal infiltration and polarization of pro-inflammatory M1 macrophages during fetal hypoxia is mitigated by fetuin-A** was added to the manuscript:

Another important cell type that contributes to the progression of renal injury are macrophages^{34, 35}. During its development, the kidney is first populated by embryo-derived, long-lived, self-renewing F4/80^{hi}CD11b^{low} cells, which maintain a resident population of macrophages³⁶. This group is complemented by bone-marrow derived circulatory F4/80^{low}CD11b^{hi} macrophages, which infiltrate and patrol the kidney, but rarely colonize the kidney except during renal injury³⁷. Besides this classification, macrophages can be further categorized according to their polarization into pro-inflammatory M1 or anti-inflammatory M2 cells. We used a FACS approach to characterize the macrophage population in E18.5 hypoxic or normoxic fetal kidneys. Strikingly, in hypoxia the composition of renal macrophages was shifted towards infiltrating F4/80^{low}CD11b^{hi} cells, partially replacing the resident population (Figs. 7a-f). Furthermore, whereas most macrophages were M2 polarized (CD206⁺) in normoxic kidneys, the majority of macrophages isolated from hypoxic kidneys had adapted a M1, pro-inflammatory phenotype (Figs. 7g-k and Supplementary Fig. 9). Importantly, the fraction of M1 polarized macrophages was even more prominent in hypoxic *Ahsg* KO samples (Table 1). Taken together, we show that hypoxia promotes the infiltration and polarization of pro-inflammatory M1 macrophages (CD206⁻) in the kidney, suggesting that fetuin-A was associated with an overall anti-inflammatory milieu.

Results are shown in novel Figure 7, Table 1, and Supplementary Figure 9

Novel Figure 7

Novel Supplementary Figure 9

Markers	Normoxia			Hypoxia		
	wt	KO	P-Value	wt	KO	P-Value
CD11c ⁺	1256 ± 214	1589 ± 201	0.138	2209 ± 325	2988 ± 208	0.046
CD68 ⁺	48 ± 11	58 ± 4	0.245	121 ± 21	167 ± 9	0.047
CD80 ⁺	265 ± 61	357 ± 51	0.132	442 ± 42	674 ± 31	0.003
CD86 ⁺	796 ± 113	1037 ± 93	0.052	2044 ± 387	2503 ± 219	0.218

Table 1. Number of macrophages expression markers of M1 polarization per 500,000 cells.

The discussion was amended as follows:

We further show that hypoxia induced a shift from anti-inflammatory M2 to pro-inflammatory M1 macrophages, which was even more prominent in fetuin-A KO mice. The polarization from M2 to M1 could be directly mediated by the low availability of oxygen during fetal hypoxia, since M1 macrophages rely on glycolysis to obtain energy, whereas M2 macrophages make use of oxidative metabolic processes⁷⁶. The reduction in M2 macrophages might directly impair nephrogenesis in hypoxia, given the trophic function of tissue resident F4/80^{hi}CD206⁺ M2 macrophages in the developing kidney^{77,78}.

Reviewer #3 (Remarks to the Author):

The authors very well addressed my comments made at first revision.

We thank Reviewer #3 again for his comments.

Reviewer #4 (Remarks to the Author):

The authors use a well-established murine model of hypoxia-induced intrauterine growth restriction (IUGR) to characterize the effect of impaired fetal growth on the kidney. Using a fetuin-A (Ahsg) KO model, the experiments described demonstrate that fetuin-A is an HIF target gene and protects against CDK in IUGR cases.

For clarity to the authors: Given that I did not review the first submission of this paper, I focused my attention on the responsiveness of the authors to initial concerns raised by the reviewers and read through the manuscript to identify any additional issues. In my view, the authors were extremely thorough in their responsiveness to reviewer concerns and made substantial improvements to their manuscript. One of my predominant concerns would have been the effect of hypoxia to reduce maternal caloric intake on the primary outcomes of interest, and that was sufficiently addressed by the authors.

We thank Reviewer #4 for his positive comments on our response to the points raised by former Reviewer #2.

A few other comparatively minor issues noted:

Line 40: LBW is traditionally defined as a birth weight < 2500g rather than < the 10th percentile. LBW is not always a consequence of IUGR (i.e., there are lots of reasons an infant can be born LBW, including non-pathological causes).

We agree with Reviewer #4 and have corrected this point in the text:

LBW, defined by the WHO as a birthweight below 2,500 g regardless of gestational age is either caused by being small for gestational age (SGA), preterm birth or a combination of thereof. SGA, most commonly a consequence of intrauterine growth restriction (IUGR), is a condition in which a fetus does not reach its genetically pre-determined growth potential. It is defined as a birth weight below the 10th percentile, or more stringent as a weight 2 SD below the mean³.

Line 57: "Population studies conducted among highland dwellers of various ethnicities found a clear correlation between altitude and incidence of renal disease making high altitude pregnancy the most important cause of maternofetal hypoxia and IUGR worldwide" The connection between the two sections of this sentence is unclear. (i.e., Why does a correlation between altitude and renal disease make HA pregnancy the most important cause of maternofetal hypoxia?)

We have partially rewritten the introduction and corrected this unclear connection.

Line 61: Use of the word proof is too strong. Suggest “robust mechanistic evidence to support Barker’s hypothesis” or something similar.

We have substituted the word proof accordingly.

Did authors consider fetal:placental ratio as an index of placental insufficiency?

Placental insufficiency is defined as how many grams of fetus are produced per gram of placenta, denoted as the fetal:placental ratio (F:P). In mice, the F:P increases with gestational age, reflecting the accelerated growth rate of the embryo in comparison to the placenta at the end of gestation. Placental weight reaches a maximum by E16.5, whereas fetal weight increases exponentially between E12.5 and E18.5 [13]. Furthermore, F:P values in mice differ among strains, laboratories and depend on many environmental conditions, making general conclusions based on the obtained values difficult to interpret, unlike in humans where certain percentiles have been defined [14]. Nevertheless, we have calculated the F:P for our E18.5 normoxic (8.55), hypoxic (7.27) and caloric control fetuses (8.86) using bulk data, since we could not evaluate individual value pairs. These values are within range of other published F:P in mice [15], and thus do not give evidence for any relevant placental insufficiency.

Line 111: This is the first mention that a KO model was used and therefore seems to pop out of nowhere. The authors should include some information in the abstract and introduction re: the KO model used for the experiments. I found both the abstract and the introduction to be too lean, lacking detail about experiments/models (abstract) and background (introduction).

We have considered this criticism and partially rewritten the abstract and introduction to include more details about the animal models, experiments and background, However, due to formatting constraints imposed by the journal space is limited.

References

1. *Reproducibility: let's get it right from the start.* Nat Commun, 2018. **9**(1): p. 3716.
2. Wilkinson, L.J., et al., *Renal developmental defects resulting from in utero hypoxia are associated with suppression of ureteric β -catenin signaling.* Kidney Int, 2015. **87**(5): p. 975-83.
3. Gonzalez-Rodriguez, P., et al., *Fetal hypoxia results in programming of aberrant angiotensin ii receptor expression patterns and kidney development.* Int J Med Sci, 2013. **10**(5): p. 532-8.
4. Walton, S.L., et al., *Late gestational hypoxia and a postnatal high salt diet programs endothelial dysfunction and arterial stiffness in adult mouse offspring.* J Physiol, 2016. **594**(5): p. 1451-63.
5. Walton, S.L., et al., *Prolonged prenatal hypoxia selectively disrupts collecting duct patterning and postnatal function in male mouse offspring.* J Physiol, 2018. **596**(23): p. 5873-5889.
6. Fathollahipour, S., P.S. Patil, and N.D. Leipzig, *Oxygen Regulation in Development: Lessons from Embryogenesis towards Tissue Engineering.* Cells Tissues Organs, 2018. **205**(5-6): p. 350-371.
7. Hemker, S.L., S. Sims-Lucas, and J. Ho, *Role of hypoxia during nephrogenesis.* Pediatr Nephrol, 2016. **31**(10): p. 1571-7.
8. Schulz, C., et al., *A lineage of myeloid cells independent of Myb and hematopoietic stem cells.* Science, 2012. **336**(6077): p. 86-90.
9. Munro, D.A.D. and J. Hughes, *The Origins and Functions of Tissue-Resident Macrophages in Kidney Development.* Front Physiol, 2017. **8**: p. 837.
10. Galván-Peña, S. and L.A. O'Neill, *Metabolic reprogramming in macrophage polarization.* Front Immunol, 2014. **5**: p. 420.
11. Rae, F., et al., *Characterisation and trophic functions of murine embryonic macrophages based upon the use of a Csf1r-EGFP transgene reporter.* Dev Biol, 2007. **308**(1): p. 232-46.
12. Muthukrishnan, S.D., et al., *Nephron progenitor cell death elicits a limited compensatory response associated with interstitial expansion in the neonatal kidney.* Dis Model Mech, 2018. **11**(1).
13. Coan, P.M., A.C. Ferguson-Smith, and G.J. Burton, *Developmental dynamics of the definitive mouse placenta assessed by stereology.* Biol Reprod, 2004. **70**(6): p. 1806-13.
14. Wallace, J.M., S. Bhattacharya, and G.W. Horgan, *Gestational age, gender and parity specific centile charts for placental weight for singleton deliveries in Aberdeen, UK.* Placenta, 2013. **34**(3): p. 269-74.
15. Lean, S.C., et al., *Placental Dysfunction Underlies Increased Risk of Fetal Growth Restriction and Stillbirth in Advanced Maternal Age Women.* Sci Rep, 2017. **7**(1): p. 9677.

REVIEWERS' COMMENTS

Reviewer #1 (Remarks to the Author):

Comments on the Ms entitled "Fetuin-A, a novel HIF target, safeguards tissue integrity during hypoxic stress" (NCOMMS-19-15730)

In the revised Ms the authors have clarified many points raised for revision with additional experiments and incorporated the results in the text accordingly. These modifications improved the quality of Ms considerably. However, a few important aspects require authors' attention which are provided below.

Authors have demonstrated that hypoxia aggravate the infiltration of macrophage and their polarization to pro-inflammatory phenotype or M1 in fetal kidney and this phenomenon gets further augmented in FetuinA KO mice which indicate FetuinA could reduce macrophage infiltration and polarization to pro inflammatory macrophage in hypoxic fetal kidney. At this point, at least a basic information regarding how FetuinA could prevent these is indeed very important. Although the authors have mentioned that amount of FetuinA in fetal kidney is 60 fold lower than the secreted amount from the liver, but the exact amount has not been provided. It would have been reasonable to quantify the differences and that would clarify the situation better. Moreover, when such a substantial lower amount of FetuinA in fetal kidney could produce a significant protection, it is presumable that there is a possibility of participation of cellular receptor(s) in FetuinA's protective activity; obviously, exploration of this would be interesting. I hope authors will consider the above mentioned points.

Reviewer #4 (Remarks to the Author):

All of my comments have been sufficiently addressed.

Point-to-point rebuttal to the Reviewer's comments

We have addressed the comments of Reviewer #1 point by point and included their discussion in the last revision of our manuscript.

Reviewer #1 (Remarks to the Author):

Comments on the Ms entitled “Fetuin-A, a novel HIF target, safeguards tissue integrity during hypoxic stress”(NCOMMS-19-15730)

In the revised Ms the authors have clarified many points raised for revision with additional experiments and incorporated the results in the text accordingly. These modifications improved the quality of Ms considerably. However, a few important aspects require authors' attention which are provided below.

Authors have demonstrated that hypoxia aggravate the infiltration of macrophage and their polarization to pro-inflammatory phenotype or M1 in fetal kidney and this phenomenon gets further augmented in FetuinA KO mice which indicate FetuinA could reduce macrophage infiltration and polarization to pro inflammatory macrophage in hypoxic fetal kidney. At this point, at least a basic information regarding how FetuinA could prevent these is indeed very important.

We think that Reviewer #1 is specifically referring to article “Adipocyte Fetuin-A Contributes to Macrophage Migration into Adipose Tissue and Polarization of Macrophages” by Chatterjee et al. in 2013 (DOI 10.1074/jbc.C113.495473), which attributed fetuin-A a promoting role for the polarization of M1 pro-inflammatory macrophages, and is disturbed by the fact that we did not reproduce these results. Indeed, our experiments clearly show that fetuin-A reduces macrophage infiltration and M2 to M1 polarization. However, we believe that our results do not invalidate in any way these previous findings, but on the contrary strengthen the hypothesis, that fetuin-A plays an important role in modulating macrophage responses. Thus both observations reflect 2 sides of the same coin, the already known Janus nature of this protein, highlighting similarities: the ectopic (non-hepatic) expression of fetuin-A, but also disparities, namely different subsequent signaling events and phenotypic outcomes. Unquestionable, the ectopic expression of fetuin-A exerts an influence on the behavior of tissue macrophages: in adipose tissue it contributes to a pro-inflammatory polarization (reviewed by Trepanowski et al., 2014, doi:10.1038/ijo.2014.203), whereas in hypoxic kidneys it keeps the polarization of M1 macrophages in check. And exactly here lies the crux of the matter: the involved tissues, the upstream mediators of fetuin-A expression, and the downstream task of fetuin-A. On the one hand, excess lipids in adipose tissue due to obesity stimulate the local production of fetuin-A via the TLR4 and Nf- κ B signaling cascade, creating a local microenvironment that stimulates M1 polarization and the release of proinflammatory cytokines along the same signaling axis (Pal et al., 2012, doi:10.1038/nm.2851; Chatterjee et al., 2013, DOI 10.1074/jbc.C113.495473). Co-authors of these papers also stated that: “Such interactions have only been reported to take place in the adipose tissue...” (Mukhopadhyay and Bhattacharya, 2016, DOI 10.1007/s00125-016-3866-y). Hence, these events seem to be the preferred signaling pathway in this setting. On the other hand, in our case, ectopic fetuin-A expression is downstream of HIF signaling triggered by severe hypoxia, for which the kidney is especially vulnerable. In this different

environment, the downstream role of fetuin-A is also very different: by binding excess calcium that is released from damaged or apoptotic cells and the formation of CPPs, fetuin-A reduces mineral stress in macrophages, thus protecting against some of the pro-inflammatory and harmful effects that are emanating from calcium phosphate nanocrystals (Smith et al., 2013, doi:10.1371/journal.pone.0060904). This function of fetuin-A does not seem to be mediated via TLR4, but involves the scavenger receptor A (SRA) system (Smith et al., 2013, doi:10.1371/journal.pone.0060904; Herrmann et al., 2012, DOI: 10.1161/CIRCRESAHA.111.261479). In line with this, it was shown by several groups that Fetuin-A indeed reduced the expression of pro-inflammatory cytokines (Smith et al., 2013, doi:10.1371/journal.pone.0060904; Dziegielewska et al., 1997, doi.org/10.1016/S0165-2478(97)00126-0). These latter aspects were already mentioned in our previous manuscripts. Summarizing this part, we think that in adipose tissue fetuin-A acts as an endogenous ligand for TLR4 through which lipids induce a pro-inflammatory response (fetuin-A as “bad guy”), while in hypoxic kidneys locally produced fetuin-A boosts the clearance of calcium nanocrystals, thus diminishing the pro-inflammatory cascade downstream of excessive calcium release (fetuin-A as “good guy”) This occurs - as previously stated by reviewer #1 - without the presence of excess lipid.

Although the authors have mentioned that amount of FetuinA in fetal kidney is 60 fold lower than the secreted amount from the liver, but the exact amount has not been provided. It would have been reasonable to quantify the differences and that would clarify the situation better. Moreover, when such a substantial lower amount of FetuinA in fetal kidney could produce a significant protection, it is presumable that there is a possibility of participation of cellular receptor(s) in FetuinA's protective activity; obviously, exploration of this would be interesting. I hope authors will consider the above mentioned points.

We believe that comparing the concentration of circulatory fetuin-A secreted by the liver [g/liter plasma] with the concentration of fetuin-A in the renal tissue [g/g tissue] would convey little additional useful information. With all due respect, this would be as comparing apples and oranges. Our results clearly show that the mRNA expression level of fetuin-A in the kidney is 60-fold lower than that in the liver. However, the liver is releasing fetuin-A for systemic use, distributed via the circulation, whereas fetuin-A produced in the kidney plays a locally confined role and has no systemic relevance. As the liver is 4-fold larger than the kidneys, we believe that the concentration of locally produced fetuin-A in the kidneys is sufficiently high to produce a significant renal protection. A feasible alternative to determine renal fetuin-A protein production would be to quantify the amount of fetuin-A released into the medium from primary cultures of proximal tubule cells, a method that has been previously used to estimate the local production of fetuin-A in adipocytes (Chatterjee et al., 2013, DOI 10.1074/jbc.C113.495473). Although such a cell culture experiment could indeed provide us with an exact amount, it remains just a number which we do not believe to be very useful outside a physiological context, considering our extensive in vivo results.

Regarding the possibility of participation of cellular receptor(s) in the protective activity of locally produced fetuin-A, we think that Reviewer #1 again is referring to the studies from Dr. Bhattacharya's group (Pal et al., 2012 doi:10.1038/nm.2851 and Chatterjee et al., 2013, DOI 10.1074/jbc.C113.495473) where fetuin-A is acting via TLR4. Here, in a different setting, the mode of action is based on the ability of fetuin-A to scavenge calcium nanocrystals, which neutralizes their pro-inflammatory features. At this step no cellular receptors are

required. However, as mentioned above, receptors like SRA on macrophages come into play to definitely remove the fetuin-A calcium particles from the site of action, as previously shown by our collaborative group (Herrmann et al., 2012, DOI: 10.1161/CIRCRESAHA.111.261479).

According to the points discussed here and above, we have made several adjustments in the last version of our manuscript, which are highlighted in green in the text:

A previous study by Chatterjee and colleagues has shown that fetuin-A promoted the polarization of M1 pro-inflammatory macrophages in adipose tissue⁸⁵, whereas we report here that fetuin-A reduces macrophage infiltration and M2 to M1 polarization. Our results do not invalidate in any way these previous findings, but on the contrary, strengthen the hypothesis that fetuin-A plays an important role in modulating macrophage responses. Both observations reflect the 2 sides of the same coin, the already known Janus nature of this protein, highlighting similarities: the ectopic (non-hepatic) expression of fetuin-A, but also disparities, namely different subsequent signaling events and phenotypic outcomes. On the one hand, excess lipids in adipose tissue due to obesity stimulate the local production of fetuin-A via the TLR4 and Nf-kB signaling cascade, creating a local microenvironment that stimulates M1 polarization and the release of proinflammatory cytokines along the same signaling axis⁸⁵⁻⁸⁷. Yet, such interactions have only been reported to take place in the adipose tissue⁸⁸. On the other hand, in our case ectopic fetuin-A expression is downstream of HIF signaling triggered by severe hypoxia, for which the kidney is especially vulnerable. The downstream role of fetuin-A is here also very different: by binding excess calcium that is released from damaged or apoptotic cells and the formation of CPPs, fetuin-A reduces mineral stress in macrophages, thus protecting against some of the pro-inflammatory and harmful effects that are emanating from calcium phosphate nanocrystals⁵⁴. This function of fetuin-A does not seem to be mediated via TLR4, but involves the scavenger receptor A (SRA) system^{54, 70}.

Unlike mRNA, exact quantification of fetuin-A protein produced locally in the kidney is not trivial due to the uptake of filtered hepatic fetuin-A into the cells of the proximal tubules, which cannot be easily distinguished from fetuin-A of renal origin. However, given the 4-fold size difference between liver and kidneys and the role that these 2 organs play in fetuin-A distribution (the liver is releasing fetuin-A into the circulation for systemic use, whereas the fetuin-A produced in the kidney upon hypoxic injury plays a local role with no systemic relevance), we believe that the concentration of locally produced fetuin-A in the kidneys is sufficiently high to produce a significant renal protection.

Reviewer #4 (Remarks to the Author):

All of my comments have been sufficiently addressed.

Thank you very much !